# Comparing Remote and Proximal Sensing of Agrometeorological Parameters across Different Agricultural Regions in Croatia: A Case Study Using ERA5-Land, Agri4Cast, and In Situ Stations during the Period 2019–2021

Dora Kreković [1,*], Vlatko Galić [2], Krunoslav Tržec [1], Ivana Podnar Žarko [1] and Mario Kušek [1]

[1] Internet of Things Laboratory, Faculty of Electrical Engineering and Computing, University of Zagreb, HR-10000 Zagreb, Croatia; krunoslav.trzec@fer.hr (K.T.); ivana.podnar@fer.hr (I.P.Ž.); mario.kusek@fer.hr (M.K.)
[2] Agricultural Institute Osijek, HR-31000 Osijek, Croatia; vlatko.galic@poljinos.hr
* Correspondence: dora.krekovic@fer.hr

**Abstract:** The paper evaluates the usability of remote satellite-based and proximal ground-based agrometeorological data sources for precision agriculture and crop production in Croatia. The compared agrometeorological datasets stem from the open-access data sources Copernicus CDS and the Agri4Cast portal, and commercial in situ agrometeorological stations (PinovaMeteo) which monitor environmental parameters relevant to the physiological state of crops. The study compares relevant parameters for 10 different locations in Croatia for three consecutive years (2019, 2020, and 2021) to investigate whether model-based data from ERA5-Land and Agri4Cast are well-correlated with ground measurements from independent in situ stations (PinovaMeteo) for specific agrometeorological parameters (air and soil temperature, and precipitation). Our results indicate the following: both the ERA5-Land and Agri4Cast datasets show mostly strong positive correlations with ground observations for air temperature, modest correlations for soil temperature, but modest or even low correlations for precipitation. Analysis of the residuals indicates higher overall residual values, especially in areas with complex topography and near large bodies of water or the sea, and deviations of residuals that may limit the usability of satellite- and model-based data for decision-making in agriculture.

**Keywords:** remote sensing; ground-based sensing; ERA5-Land; Agri4Cast; precision agriculture; Pearson correlation; principal component analysis

## 1. Introduction

Precision agriculture (PA) is a data-driven approach to farming that uses a range of ICT solutions—remote sensing, Internet of Things (IoT), artificial intelligence (AI)—to improve and increase crop yields and the profitability of agricultural production, while reducing the amount of resources needed for food production, such as the amount of water, fertilizers, herbicides, and insecticides [1,2]. This blend of technologies allows accurate provision and analysis of field data in (near) real time, thus automating production and decision-making processes to maximize profits and increase crop productivity while promoting environmental sustainability [3]. The major difference compared to the classical approach to agricultural production is that PA allows decisions to be made quickly and at different levels of spatial granularity, ranging from an entire field to a square meter of field, to account for the spatial variability of crops. Such decisions are facilitated by monitoring both crops and fields through different sensing technologies applied either in close proximity to the crops (*proximal ground-based sensing*) or remotely from the air/sky (*remote air- or space-borne sensing*) [4].

Proximal ground-based sensing monitors environmental parameters and the status of crops in fields at close range. Devices for proximal sensing may be (i) hand-held,

(ii) mounted on tractors or farm machinery, or (iii) stationary [2]. Stationary devices are usually in situ agrometeorological stations equipped with various sensors for monitoring environmental parameters (air and soil temperature/humidity, precipitation levels, wind speed, solar and global radiation, and evapotranspiration) and are among the most popular ground-based solutions. They fall into the category of IoT solutions since agrometeorological stations are connected to the Internet and back-end platforms to provide continuous access to acquired sensor data in (near) real time [5]. Remote space-borne sensing involves using satellites to collect data about crops and fields from space, while airborne sensing utilizes unmanned aerial vehicles (UAVs) for mobile monitoring at much shorter distances. Both UAVs and satellites typically use cameras (RGB, multispectral or hyperspectral) to intermittently collect the images required to determine vegetation indices to assess the physiological state of crops or the characteristics of soil [4]. The largest difference between these three sensing options is in the effort required to deploy the sensors and to start collecting data. More importantly, the employed sensors differ based on the spatial (size of the pixel that represents an area on the ground), spectral (number of captured bands), and temporal (interval between two consecutive measurements) resolution they provide [6].

Remote space-borne and in situ sensing is a mature technology which has led to the creation of open datasets estimating a wide range of land surface parameters at high resolution that are very useful for PA. Satellite data complemented by ground measurements serve as the basis for the ERA5-Land reanalysis dataset that provides information on land surface parameters [7]. The ERA5-Land dataset is produced by the European Centre for Medium-Range Weather Forecasts (ECMWF) as part of the Copernicus Climate Change Service (C3S) and has been offered openly since 2019. It provides hourly estimates of a wide range of land surface parameters at a spatial resolution of 9 km. Estimates are produced by combining historical observations with a numerical weather prediction (NWP) model. The second comparable data source stemming primarily from in situ sensing is Agri4Cast provided by the EU Joint Research Centre (JRC) [8], which focuses on crop monitoring and yield forecasting at the European level. Agri4Cast is a specialized product for PA, which, among other services, releases datasets for a variety of agrometeorological data, such as air and soil temperature, precipitation, and evapotranspiration. In this work, we use the Agri4Cast agrometeorological dataset, which contains meteorological observations from weather stations interpolated on a $25 \times 25$ km grid as average daily values.

In this paper, we analyze and compare two open-access datasets that provide estimates of land surface parameters based on remote space-borne and in situ sensing: ERA5-Land and Agri4Cast. We evaluate how well their models match the observations obtained from independent ground-based in situ agrometeorological stations (PinovaMeteo) deployed in different farming regions of Croatia. Measurement data from 10 different locations in Croatia for three consecutive years (2019, 2020, and 2021) are used in our study to investigate the following environmental parameters: air and soil temperature, and precipitation. A statistical analysis is performed to compare PinovaMeteo readings with the corresponding ERA5-Land and Agri4Cast datasets to determine the statistical significance of the similarity between model-based parameter estimates with a gridded resolution and actual in situ observations from a micro-location.

In particular, our original contribution includes the following:

- We use the Pearson correlation coefficient and significance test to assess the correlation of the ERA5-Land and Agri4Cast datasets with the available ground measurements for the listed agrometeorological parameters at 10 different locations in Croatia.
- We perform principal component analysis to evaluate the residuals between the different data sources and deviations of remote sensing compared to actual ground-based observations to detect potential systematic errors.

A number of studies have compared satellite-based agrometeorological data with ground measurements from on-site stations, as we discuss in detail in Section 2. However, it is increasingly important to validate the accuracy of remote sensing models against ground measurements since satellite-based data are becoming important for monitoring

and managing agricultural production. In our previous analysis [9], we compared a larger number of agrometeorological parameters from the same three data sources for a single location in Croatia using only the Pearson correlation coefficient. In this paper, we extend the statistical analysis and compare a significantly larger dataset covering 10 locations in Croatia with different topographical features and in different climatic regions for the period from January 2019 to December 2021.

In a broader context, our aim is to determine the extent of usefulness and limitations of remote sensing products compared to ground-based observations for PA. Although remote sensing is limited in terms of spatial and temporal resolution compared to ground-based sensing, it is widely used by various stakeholders in agribusiness due to its global coverage and cost-effectiveness. Typical applications include crop yield prediction, water management, and disaster risk assessment. However, micro-meteorological conditions often affect farmer's fields and require rapid data-driven decision making. In such cases, the ground-based stations show unprecedented temporal resolution, however, with a strong limitation in spatial context as they are bound to a specific micro-location. Recent advances in computational tools, especially in the areas of IoT technologies and edge computing, require the integration of multiple data sources to unify the model-based monitoring with ground-based stations in a robust, scalable manner. For this effort to succeed, comparison of data sources over different topological and geographic scenarios is needed. Our research thus contributes to the first step in this integration, comparing time-series data from three data-sources, aiming to capture their weaknesses and advantages for integration into the cross-calibrated agrometeorological system for Agriculture 4.0.

The rest of the paper is organized as follows: Section 2 presents related work investigating existing studies which compare satellite-based agrometeorological datasets with ground-based observations from in situ stations. The data sources in Croatia used for our statistical analysis are described in Section 3, while the results of our statistical analysis and comparison are presented in Section 4 for 10 chosen locations in different regions of the country. Section 5 discusses the results of our study, while Section 6 concludes the paper.

## 2. Related Work

Remote sensing and the potential of satellite data to transform agricultural practices towards eco-efficiency and higher productivity has been highlighted in a number of studies: It can support decision making in an accessible and effective manner with a large spatial coverage and relatively low cost [10]. The use of remote sensing technologies can be seen in many PA applications today, e.g., crop monitoring, irrigation management, nutrient application, disease and pest management, and yield prediction [2].

There are a number of studies comparing satellite agrometeorological data with ground measurements from in situ stations focusing either on a specific land parameter [11,12] or region [12–14]. A recent study that is the most relevant to our work compared the ERA5 and ERA5-Land datasets with ground-based agrometeorological data from 66 automatic weather stations in Italy [14]. The study explored the following parameters: solar radiation, air temperature, relative humidity, wind speed, and reference evapotranspiration. The main findings were the following: The air temperature estimates offered the most accurate reanalysis predictions, while reference evapotranspiration estimates were assessed as reliable. The authors point out that the climatic conditions affected the accuracy of the reanalysis products. In contrast to previous studies which have identified topographic features to affect the accuracy of remote sensing products (e.g., [15]), this study did not confirm such a relationship.

The difficulties in accurately measuring and forecasting agrometeorological parameters are particularly acute in the case of precipitation. This is because precipitation is inherently heterogeneous in space and time. Therefore, the authors in [12] performed a study aimed at a comparative analysis of satellite-based rainfall products (SRPs) and gauge data to ascertain the reliability of using SRPs for daily rainfall measurements in Zambia. SRPs were compared to rain gauge data from 35 meteorological, agrometeorological, and climatological stations in Zambia for the period 1998–2015. Statistical analyses were conducted at different temporal

scales (e.g., daily, monthly, seasonal, annual). The study showed that the use of carefully validated SRPs was suitable as a substitute for daily rainfall measurements in Zambia. While the results of the work are useful for Zambia, they cannot be generalized to other regions as the behavior of SRPs differs from one region to the other [16].

Researchers have recently pointed out the challenges and limitations associated with both remote and proximal sensing [4]. On the one hand, global coverage and cost-effectiveness are identified as the main advantages of remote sensing, while it has limitations in terms of spatial and temporal resolution. On the other hand, the low spatial distribution and high cost of ground measuring stations limit their monitoring coverage for large agricultural areas, while more accurate measurements and dense temporal resolution are their major strengths. Recent works are, thus, exploring the complementary use of remote and proximal sensing to provide more accurate agrometeorological data for PA practices through sensor fusion.

For example, in [17], the authors proposed simultaneous use of ground stations and satellite data to improve and enhance agrometeorological products. They presented examples of the use of meteorological products combining classical ground measurements and data from meteorological radars and satellites, applied in an agrometeorological service provided by the Institute of Meteorology and Water Management in Poland. It was emphasized that further improvement in the methods for sharing agrometeorological data and combining data from ground stations with increasingly better satellite products are essential for modern agriculture in the conditions of progressive climate change. Another example is a data fusion combining remote sensing, UAVs, and autonomous driving machines to optimise vineyard cultivation and production [18].

The authors of [19] used Landsat 8 satellite images together with a net of agrometeorological stations data for acquiring the surface temperature in the northwestern side of São Paulo state, Brazil. The authors performed a performance evaluation of the methods suitable for acquiring surface temperature by using high-resolution satellite images without a thermal band, having available spatially distributed weather data.

Satellite image time series (SITS), such as those obtained by Sentinel-2 (S2) satellites, provide a large amount of information due to their combined temporal, spatial, and spectral resolutions. The high revisit frequency and spatial resolution of S2 result in the availability of detailed information for analyzing small objects and increase in the probability of acquiring cloud-free images. These characteristics are of interest in precision agriculture where temporally dense SITS can benefit the understanding of crop behaviors. Therefore, the authors of [20] proposed a method suitable for the analysis of small crop fields in S2 dense SITS which could account for the S2 characteristics. The method fuses spatiotemporal information, analyzes data spatiotemporal evolution, and extracts relevant spatiotemporal information. The effectiveness of the proposed method was corroborated by experiments carried out on S2-SITS acquired over an area located in Barrax, Spain.

## 3. Available Agrometeorological Data Sources in Croatia

Croatia consists of three main geographical regions [21]: the Pannonian and para-Pannonian plains in the north and north-east, the central mountain belt in the west and south, and the Croatian coastal area. The Pannonian plains are the most fertile agricultural regions in Croatia, enriched by alluvial deposits from the Sava and Drava rivers. The central mountain belt offers some arable, meadow, and pasture land, while the coastal region is mostly barren and mountainous with little agricultural land.

Croatia has a moderately warm and rainy climate with monthly average temperatures ranging from $-3\,°C$ (in January) to $18\,°C$ (in July), depending on the season and region. The warmest areas are found on the Adriatic coast and its immediate hinterland, characterized by a Mediterranean climate. The coldest parts of the country are the mountainous central regions, Lika and Gorski Kotar, where a snowy forest climate prevails at altitudes above 1200 m. The average annual precipitation ranges from 300 to 3500 mm, depending on the geographical region and climate type. The prevailing winds are determined by the

local conditions. The sunniest parts of the country are the outer southern islands, Hvar and Korčula.

Our study includes a total of 10 sites with agrometeorological stations distributed throughout Croatia, as shown in Figure 1. Four sites are in the Slavonia region (Suhopolje, Skenderovci, Našice, Kamenac), one in Međimurje (Nedelišće), one in Istria (Funtane), one in Lika (Otočac), and three in Dalmatia (Dugi Otok, Oklaj, Potomje).

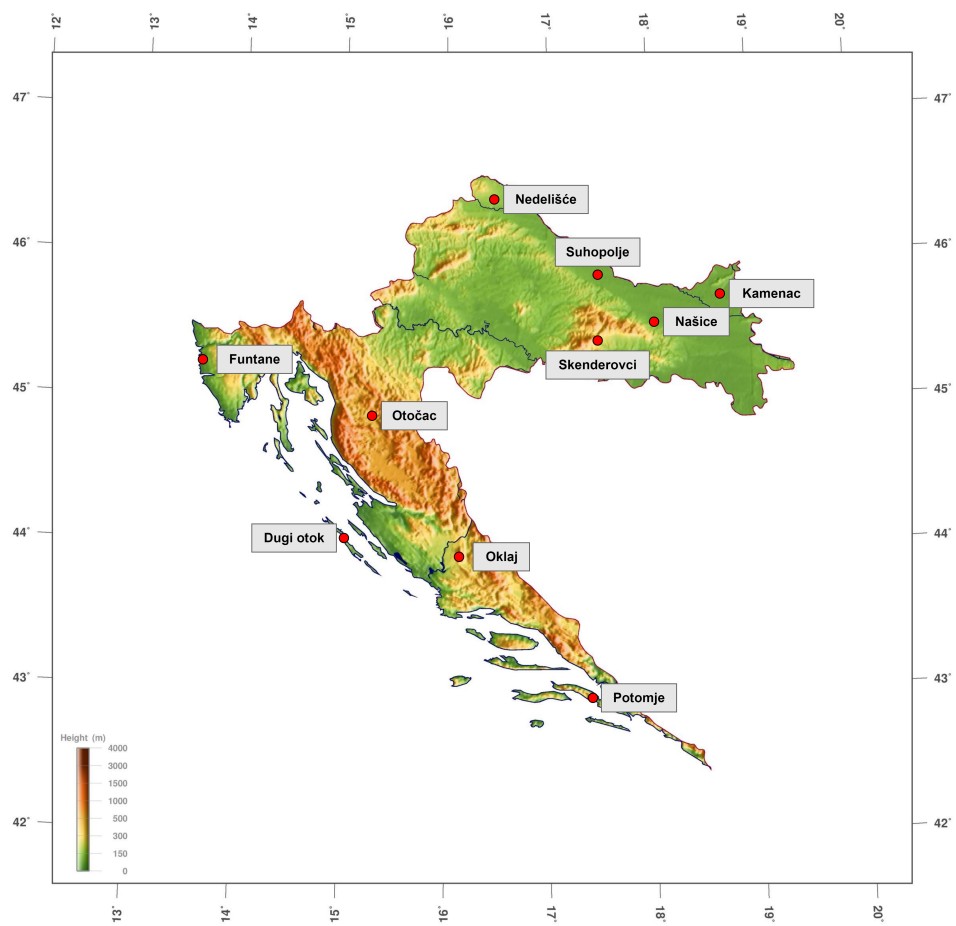

**Figure 1.** Map of Croatia with stations highlighted by gray squares

### 3.1. Copernicus Climate Data Store and ERA5-Land Dataset

The Copernicus services mainly rely on data from the Sentinel satellites, which are owned by the European Union and used primarily for Earth observation. The first satellite was launched in 2014. Additionally, some of the data are obtained from ground-based meteorological stations, ocean buoys, and air quality sensors [22]. These in situ measured data are used to calibrate and verify the satellite data and to provide reliable and consistent information.

The Copernicus climate data archive includes the ERA5 dataset, a fifth-generation global atmospheric reanalysis from the European Centre for Medium-Range Weather Forecasts (ECMWF). Covering the period from January 1950 to the present, ERA5 serves as a comprehensive resource of Earth observation. The dataset utilizes a more advanced version of the ECMWF Integrated Forecast System model, offering increased temporal output and higher horizontal and vertical resolutions [23].

The data used in the study are from the ERA5-Land dataset. Compared to ERA5 and the old ERA-Interim, the main advantage of ERA5-Land is the improved horizontal resolution, which is 9 km compared to 31 km (ERA5) or 80 km (ERA-Interim), while the temporal resolution is hourly, as in ERA5. ERA5-Land provides higher resolution data

based on an additional examination of the land component of the ECMWF ERA5 climate reanalysis [7,24]. This reanalysis method seamlessly integrates observations and models to fill data gaps, contributing valuable insights about global weather and climate patterns.

The ERA5-Land dataset relies on atmospheric forcing derived from ERA5 near-surface meteorology state and flux fields [7]. Parameters such as air temperature, specific humidity, wind speed, surface pressure, and surface fluxes, including radiation and precipitation, are crucial parameters of the ERA5-Land dataset. A careful interpolation process is applied to transform these parameters from the ERA5 resolution to the ERA5-Land resolution, with precision improved by a linear interpolation method based on a triangular mesh. The hourly atmospheric forcing for ERA5-Land is maintained consistently over its entire production period. This is achieved by assimilating conventional meteorological and satellite observations using a four-dimensional variational assimilation system (4D-Var) and simplified extended Kalman filter (SEKF) systems. The ECMWF land surface model, specifically, the Carbon Hydrology-Tiled ECMWF Scheme for Surface Exchanges over Land (CHTESSEL) forms the core of the ERA5-Land model. Further details about the model can be explored in the Integrated Forecasting System (IFS) documentation [25].

The Climate Data Store (CDS) is the core of the Copernicus Climate Change Service, also known as C3S [26]. The CDS provides free access to information about past, present, and future climate observations, and serves as a one-stop site for users to explore climate data. It provides a variety of quality-controlled climate data that are made available to users in a consistent and dependable manner. The Copernicus site [27] emphasizes that the environmental data processed by Copernicus services are derived from Earth observation satellites and 'in situ' sensors. These sensors, whether ground-based, in the ocean, or in the air, offer precise measurements at specific sites. Notably, for CDS, satellite observations are integrated into the calculations, enhancing both the accuracy and spatial resolution of the model for ERA5-Land parameters explored by this study. CDS API is a service that provides programmatic access to CDS data in Python (using the CDS API client).

Table 1 lists the selected parameters from the ERA5-Land dataset which we used in this study. The dataset also contains other data types that were not explored in this study (For a more extensive list of comparable parameters from the three data sources, please refer to [9]); however, those mentioned were selected since they are informative for PA and were adequate for the subsequent statistical analysis. The data are retrieved through the CDS API, which is accessed via a Python script. To properly download the data, users must first register and then generate a *.cdsapirc* file.

**Table 1.** ERA5-Land parameters selected for the statistical analysis.

| Name | Unit | Description |
|---|---|---|
| 2 m temperature | K | air temperature at 2 m above the surface of land, sea, or inland waters |
| soil temperature level 1 | K | temperature of the soil in layer 1 (0–7 cm) of the ECMWF Integrated Forecasting System |
| total precipitation | m | accumulated liquid and frozen water, including rain and snow |

*3.2. Agri4Cast Portal*

The Agri4Cast data collection, developed by the Joint Research Centre (JRC) and its Monitoring Agricultural ResourceS (MARS) unit, has been an important resource of agrometeorological information to EU member states since 1975. This comprehensive dataset contains daily weather data from a network of at least 4200 weather stations, characterised by an irregular distribution and density. In addition, the dataset contains data from six weather forecast products, five of which come from ECMWF and one from the Copernicus programme. These forecast products exhibit variations in forecast depth and have a different number of possible realisations, referred to as "members". The ECMWF weather products, including the ERA model, contribute valuable forecast data to the Agri4Cast collection. Given the non-uniform distribution of weather stations and the varying distances between them, both the observed weather data and the forecast products are then interpolated onto a

fixed 25 × 25 km grid that matches the grid used by ERA and E-OBS [28]. The grid-based data acquisition assumes homogeneity within the 25 × 25 km grid cells. To achieve this, two different interpolation methods are used. Precipitation is interpolated using regression kriging, while the Crop Growth Monitoring System (CGMS) is used for the interpolation of all other meteorological elements [29].

In contrast to ERA5-Land, this dataset in not based on satellite observations, but rather on the following two components:

- daily weather information from a large synoptic weather station network, and
- six different weather forecast products, which provide a comprehensive and versatile resource for agrometeorological analyses.

Through the Agri4Cast portal, users can submit requests and download agrometeorological data for Europe. The desired data are filtered by country and then by region within each country before downloading. For the Republic of Croatia, data are provided for the continental and Adriatic regions, as well as for each county within these regions. When submitting a request, a user can specify one or more datasets to download and the desired time period. The data are available from 1 January 1979. The portal is updated once a year, and the previous year's data are usually made available in January. The Agri4Cast data are usually available 15 min after selecting the appropriate parameters and submitting the request, after which the user is notified via an e-mail address used during registration on the site. The generated data are provided as a CSV file.

The parameters from the Agri4Cast dataset used in our analysis are listed in Table 2.

**Table 2.** Agri4Cast parameters selected for the statistical analysis.

| Description | Unit |
|---|---|
| average daily temperature | °C |
| sum of precipitation per day | mm |

### 3.3. PinovaMeteo

PinovaMeteo is an in situ agrometeorological station developed by the company Pinova from Čakovec, Croatia. It is used in orchards, vineyards, vegetable gardens, plant nurseries and farms, and allows users to check current and historical measurements via mobile and web applications. The PinovaMeteo station is fully automated and connected to the Internet, so users do not need to visit it to collect data. The station is powered by a solar panel during the day and operates at night with a battery that is charged during the day. The PinovaMeteo station continuously collects data every 10 min and submits data to a backend server every half an hour or hour (more or less frequently depending on the user's needs and battery state).

The parameters measured by the PinovaMeteo agrometeorological stations which are used in our study are listed in Table 3.

**Table 3.** PinovaMeteo parameters used in the statistical analysis.

| Description | Unit |
|---|---|
| air temperature | °C |
| soil temperature | °C |
| rainfall | mm |

## 4. Comparison of Available Datasets for Locations in Croatia

To assess and compare the open access data sources, the Copernicus CDS and Agri4Cast portal, we compared the selected data types with those collected by the PinovaMeteo agrometeorological station for the listed locations in Croatia and the same time frame (January 2019 to December 2021). The data from the ERA5-Land and Agri4Cast datasets were compared with the readings obtained from the PinovaMeteo stations.

The dependence between pairs of time series is usually quantified by the Pearson correlation. In this work, we have used it to estimate the relationship between the above datasets. The Pearson correlation coefficient is a measure of overall similarity that reduces the relationship between two signals to a single value. In other words, it is defined as a measure of the linear relationship between two characteristics that expresses the ratio of the product of the covariances of $x$ and $y$ and their standard deviations. Given a pair of random variables, the formula is:

$$r = \frac{Cov(x,y)}{(\sigma_x \sigma_y)} \tag{1}$$

where:

$r$ is a Pearson correlation coefficient,

$Cov(x,y)$ is covariance of variables $x$ and $y$,

$\sigma_x$ is standard deviation of $x$,

$\sigma_y$ is standard deviation of $y$.

Correlation coefficients are calculated for comparable data types in the available data sources for each month of 2019, 2020, and 2021. The daily mean is used as input for the monthly calculation of the correlation coefficient for comparison of the PinovaMeteo readings with the Agri4Cast dataset that offers daily values, and the hourly mean to calculate the correlation between ERA5-Land and the PinovaMeteo data.

*4.1. Comparing ERA5-Land with PinovaMeteo*

The first analysis (ERA5-Land vs. PinovaMeteo) includes the following parameters: air temperature, precipitation, and soil temperature. Table 4 displays the calculated correlation values for air temperature at the nine specified locations. These values reveal the existence of dependable seasonal temperature trends, supported by the enduringly robust positive correlations observed across most months and stations during the three-year duration. To deduce these consistent seasonal temperature patterns from the correlation data, we analyze the strength, consistency, and uniformity of temperature trends among diverse months and stations throughout this extended timeframe. Positive correlations, particularly when conforming to seasonal cycles, signify that temperature variations reliably synchronize with the changing seasons. This indicates a strong agreement between the PinovaMeteo and ERA5-Land air temperature data in these periods. The observed dynamic correlation patterns, which vary from month to month, may reflect underlying temperature patterns, with lower correlations observed particularly in winter months at the beginning and end of each year. While there is no clear pattern for the months as a whole, it can be observed that some stations consistently have lower correlation values than others, namely Funtane, Potomje, and Otočac. The first two (Funtane and Potomje) are located close to the sea, while Otočac is in a valley surrounded by high mountains. It is important to consider the year-specific variations, as certain months in certain years show relatively weaker correlations. For example, January has lower correlations at certain stations in both 2019 and 2020, which could be attributed to unique weather events or local variations in these periods. In summary, these results confirm the reliability of both the PinovaMeteo and ERA5-Land datasets for monitoring temperature changes in the studied regions.

The correlation values for precipitation presented in Table 4 show that the correlation between the PinovaMeteo and ERA5-Land datasets for precipitation is consistently weak (mostly below 0.5) across different months and years, and never exceeds the threshold of 0.7. These results suggest possible discrepancies or variations in precipitation data between the two sources. There are also differences in the correlation strength between the different stations. For example, while we cannot identify a station with consistently higher correlation compared to others, stations such as Kamenac and Funtane even show negative correlations. This could indicate differences in data quality between stations. It should also

be noted that ERA5-Land covers a larger area (9 × 9 km), while Pinova measures precipitation at specific locations, which may lead to such low correlation values. The presence of negative correlations in some cases, particularly in the summer months of 2019 and 2020, raises questions about the alignment of the PinovaMeteo and ERA5-Land datasets for precipitation data during these periods. This uncertainty raises concerns about the reliability of these datasets for monitoring precipitation changes in specific scenarios.

**Table 4.** Correlations between PinovaMeteo and ERA5-Land data. Correlations below the correlation threshold value of r = 0.7 are marked in bold. The month with the strongest correlation value for each location within one year is highlighted in orange and the weakest in blue.

| Parameter | Year | Month | Stations | | | | | | | | |
|---|---|---|---|---|---|---|---|---|---|---|---|
| | | | Našice | Nedelišče | Otočac | Oklaj | Potomje | Suhopolje | Kamenac | Skenderovci | Funtane |
| Temperature | 2019 | January | 0.863 | 0.913 | 0.752 | 0.885 | 0.768 | 0.920 | 0.914 | 0.826 | 0.796 |
| | | February | 0.936 | 0.949 | 0.821 | 0.897 | 0.725 | 0.923 | 0.915 | 0.838 | 0.827 |
| | | March | 0.932 | 0.959 | 0.900 | 0.910 | 0.779 | 0.928 | 0.941 | 0.894 | 0.889 |
| | | April | 0.932 | 0.933 | 0.834 | 0.912 | 0.848 | 0.923 | 0.892 | 0.883 | 0.876 |
| | | May | 0.944 | 0.942 | 0.873 | 0.908 | 0.860 | 0.954 | 0.946 | 0.918 | 0.901 |
| | | June | 0.926 | 0.906 | 0.820 | 0.915 | 0.847 | 0.926 | 0.932 | 0.881 | 0.865 |
| | | July | 0.883 | 0.857 | 0.815 | 0.862 | 0.815 | 0.861 | 0.884 | 0.859 | 0.800 |
| | | August | 0.943 | 0.923 | 0.792 | 0.908 | 0.855 | 0.935 | 0.933 | 0.901 | 0.865 |
| | | September | 0.925 | 0.923 | 0.805 | 0.882 | 0.800 | 0.900 | 0.933 | 0.895 | 0.854 |
| | | October | 0.945 | 0.923 | 0.724 | 0.917 | 0.839 | 0.907 | 0.919 | 0.842 | 0.864 |
| | | November | 0.908 | 0.901 | 0.812 | 0.888 | 0.822 | 0.909 | 0.932 | 0.849 | 0.865 |
| | | December | 0.813 | 0.951 | 0.819 | 0.933 | 0.825 | 0.873 | 0.892 | 0.828 | 0.892 |
| | 2020 | January | 0.805 | 0.792 | **0.670** | 0.861 | **0.669** | 0.722 | 0.749 | 0.830 | 0.872 |
| | | February | 0.913 | 0.924 | 0.900 | 0.905 | 0.791 | 0.940 | 0.939 | 0.914 | 0.858 |
| | | March | 0.954 | 0.960 | 0.908 | 0.931 | 0.758 | 0.958 | 0.952 | 0.910 | 0.843 |
| | | April | 0.941 | 0.921 | 0.816 | 0.896 | 0.817 | 0.933 | 0.925 | 0.908 | 0.897 |
| | | May | 0.923 | 0.923 | 0.821 | 0.896 | 0.862 | 0.934 | 0.914 | 0.907 | 0.823 |
| | | June | 0.910 | 0.916 | 0.844 | 0.902 | 0.848 | 0.916 | 0.914 | 0.890 | 0.792 |
| | | July | 0.923 | 0.928 | 0.828 | 0.906 | 0.864 | 0.920 | 0.945 | 0.890 | 0.817 |
| | | August | 0.909 | 0.925 | 0.776 | 0.874 | 0.831 | 0.904 | 0.895 | 0.877 | 0.724 |
| | | September | 0.934 | 0.936 | 0.831 | 0.912 | 0.828 | 0.919 | 0.931 | 0.916 | 0.759 |
| | | October | 0.940 | 0.934 | 0.826 | 0.924 | 0.841 | 0.942 | 0.908 | 0.898 | 0.766 |
| | | November | 0.936 | 0.920 | 0.747 | 0.880 | 0.764 | 0.941 | 0.912 | 0.922 | 0.807 |
| | | December | 0.938 | 0.865 | 0.780 | 0.900 | 0.814 | 0.928 | 0.920 | 0.835 | 0.838 |
| | 2021 | January | 0.922 | 0.952 | 0.909 | 0.938 | 0.868 | 0.925 | 0.937 | 0.900 | 0.893 |
| | | February | 0.956 | 0.948 | 0.895 | 0.935 | 0.885 | 0.948 | 0.948 | 0.907 | 0.872 |
| | | March | 0.940 | 0.957 | 0.887 | 0.909 | 0.773 | 0.947 | 0.939 | 0.894 | 0.896 |
| | | April | 0.937 | 0.938 | 0.845 | 0.918 | 0.842 | 0.941 | 0.939 | 0.900 | 0.838 |
| | | May | 0.921 | 0.935 | 0.815 | 0.899 | 0.836 | 0.926 | 0.928 | 0.896 | 0.827 |
| | | June | 0.932 | 0.937 | 0.842 | 0.907 | 0.866 | 0.927 | 0.948 | 0.803 | 0.810 |
| | | July | 0.923 | 0.914 | 0.826 | 0.878 | 0.815 | 0.915 | 0.944 | 0.892 | 0.755 |
| | | August | 0.931 | 0.937 | 0.799 | 0.901 | 0.836 | 0.922 | 0.942 | 0.881 | 0.777 |
| | | September | 0.931 | 0.887 | 0.796 | 0.918 | 0.835 | 0.902 | 0.952 | 0.874 | 0.757 |
| | | October | 0.947 | 0.939 | 0.848 | 0.864 | **0.698** | 0.922 | 0.943 | 0.887 | 0.837 |
| | | November | 0.895 | 0.914 | **0.692** | 0.860 | 0.720 | 0.882 | 0.908 | 0.842 | 0.762 |
| | | December | 0.944 | 0.912 | 0.823 | 0.899 | 0.804 | 0.951 | 0.953 | 0.911 | 0.748 |

**Table 4.** *Cont.*

| Parameter | Year | Month | Našice | Nedelišće | Otočac | Oklaj | Potomje | Suhopolje | Kamenac | Skenderovci | Funtane |
|---|---|---|---|---|---|---|---|---|---|---|---|
| Precipitation | 2019 | January | 0.159 | 0.395 | 0.143 | 0.494 | 0.419 | 0.423 | 0.241 | 0.197 | 0.321 |
| | | February | 0.330 | 0.322 | 0.391 | 0.154 | 0.289 | 0.292 | 0.311 | 0.274 | 0.768 |
| | | March | 0.242 | 0.452 | 0.524 | 0.059 | 0.143 | 0.122 | 0.225 | 0.469 | 0.164 |
| | | April | 0.417 | 0.314 | 0.388 | 0.344 | 0.313 | 0.383 | 0.277 | 0.199 | 0.266 |
| | | May | 0.354 | 0.456 | 0.435 | 0.145 | 0.609 | 0.417 | 0.330 | 0.272 | 0.190 |
| | | June | 0.091 | 0.107 | 0.416 | 0.101 | −0.002 | 0.103 | 0.188 | 0.053 | 0.118 |
| | | July | 0.116 | 0.155 | 0.105 | 0.297 | 0.011 | 0.174 | 0.102 | 0.042 | 0.411 |
| | | August | 0.204 | 0.534 | 0.178 | 0.102 | 0.054 | 0.309 | 0.095 | 0.189 | 0.248 |
| | | September | 0.289 | 0.251 | 0.242 | 0.094 | 0.115 | 0.397 | 0.257 | 0.054 | 0.027 |
| | | October | 0.442 | 0.498 | 0.282 | 0.234 | 0.246 | 0.465 | 0.401 | 0.199 | 0.026 |
| | | November | 0.183 | 0.195 | 0.271 | 0.150 | 0.127 | 0.282 | 0.205 | 0.191 | 0.123 |
| | | December | 0.252 | 0.467 | 0.396 | 0.358 | 0.356 | 0.273 | 0.198 | 0.286 | 0.255 |
| | 2020 | January | 0.397 | 0.059 | 0.105 | 0.274 | 0.429 | 0.360 | −0.006 | 0.400 | 0.215 |
| | | February | 0.309 | 0.631 | 0.233 | 0.469 | 0.274 | 0.327 | 0.254 | 0.260 | 0.265 |
| | | March | 0.060 | 0.257 | 0.202 | 0.311 | 0.181 | 0.261 | 0.417 | 0.072 | 0.501 |
| | | April | 0.278 | 0.348 | 0.208 | 0.437 | 0.335 | 0.298 | 0.251 | 0.281 | 0.245 |
| | | May | 0.164 | 0.176 | 0.096 | 0.188 | 0.082 | 0.124 | 0.381 | 0.213 | 0.007 |
| | | June | 0.015 | 0.092 | 0.127 | 0.097 | 0.257 | 0.020 | −0.046 | 0.125 | 0.171 |
| | | July | 0.226 | 0.300 | 0.139 | 0.240 | 0.044 | 0.232 | −0.025 | 0.041 | 0.043 |
| | | August | 0.098 | 0.189 | 0.148 | 0.306 | 0.344 | 0.067 | −0.016 | 0.046 | |
| | | September | 0.239 | 0.402 | 0.353 | 0.311 | 0.146 | 0.537 | 0.093 | 0.146 | |
| | | October | 0.311 | 0.352 | 0.399 | 0.162 | 0.285 | 0.312 | 0.362 | 0.319 | |
| | | November | 0.478 | 0.516 | 0.027 | 0.479 | 0.194 | 0.251 | 0.226 | 0.383 | −0.015 |
| | | December | 0.446 | 0.457 | 0.136 | 0.287 | 0.471 | 0.577 | 0.383 | 0.469 | −0.037 |
| | 2021 | January | 0.310 | 0.341 | 0.191 | 0.318 | | 0.436 | 0.432 | 0.293 | −0.040 |
| | | February | 0.134 | 0.150 | 0.580 | 0.227 | | 0.336 | 0.226 | 0.164 | −0.024 |
| | | March | 0.600 | 0.328 | 0.074 | 0.297 | | 0.662 | 0.588 | 0.634 | −0.029 |
| | | April | 0.299 | 0.309 | 0.299 | 0.284 | | 0.383 | 0.359 | 0.300 | |
| | | May | 0.450 | 0.309 | 0.338 | 0.050 | | 0.152 | 0.357 | 0.326 | −0.009 |
| | | June | 0.064 | 0.206 | 0.697 | 0.135 | | 0.004 | 0.236 | 0.110 | 0.123 |
| | | July | 0.331 | 0.139 | 0.016 | 0.140 | | 0.149 | 0.106 | 0.268 | 0.388 |
| | | August | 0.143 | 0.341 | 0.037 | 0.229 | | 0.193 | −0.007 | 0.258 | 0.035 |
| | | September | 0.284 | 0.319 | 0.375 | 0.256 | | 0.308 | | 0.247 | 0.047 |
| | | October | 0.562 | 0.581 | 0.521 | 0.257 | | 0.379 | 0.439 | 0.445 | 0.306 |
| | | November | 0.153 | 0.186 | 0.244 | 0.149 | | 0.268 | 0.140 | 0.118 | 0.473 |
| | | December | 0.319 | 0.500 | 0.380 | 0.334 | | 0.441 | 0.278 | 0.396 | |

The following analysis is reported for five locations for which valid soil temperature readings were available. The correlation values presented in Table 5, showing the comparison of soil temperature between the PinovaMeteo and ERA5-Land datasets for the period from 2019 to 2021 at different stations, contain several noteworthy findings. They highlight the presence of seasonal variations in ground temperature patterns, with higher correlations observed in winter months, such as November and December, while the correlations are generally lower in summer months. A more distinct perspective on this trend is offered by examining the final column in Table 5, which illustrates the calculated monthly means for ground temperature correlations. The station-specific trends highlight the influence of local factors on the soil temperature data, with certain stations consistently showing stronger correlations than others. Overall, while the data suggest some degree of consistency in soil temperature patterns over the entire period studied, they also underscore the importance of

considering both seasonal variations and year-to-year influences in comprehensive analyses. For example, the Našice station has higher correlation values in 2021 than in 2020 and 2019.

**Table 5.** Correlations for soil temperature between PinovaMeteo and ERA5-Land data. Correlations below the threshold value of r = 0.7 are marked in bold. The month with the strongest correlation value for each location is highlighted in orange, the weakest in blue.

| Month | 2019 | | | | | 2020 | | | | | 2021 | | | | Mean |
|---|---|---|---|---|---|---|---|---|---|---|---|---|---|---|---|
| | Našice | Otočac | Oklaj | Potomje | Suhopolje | Našice | Otočac | Oklaj | Potomje | Suhopolje | Našice | Otočac | Oklaj | Suhopolje | |
| January | 0.748 | 0.861 | 0.718 | 0.853 | **0.620** | 0.397 | **0.594** | **0.541** | 0.584 | **0.236** | 0.895 | **0.633** | 0.918 | 0.730 | **0.664** |
| February | 0.724 | 0.730 | **0.445** | 0.812 | **0.386** | 0.731 | **0.637** | **0.496** | 0.775 | **0.457** | 0.893 | 0.767 | 0.803 | 0.760 | **0.683** |
| March | **0.674** | 0.804 | 0.427 | 0.798 | **0.287** | 0.690 | **0.544** | **0.437** | 0.701 | **0.423** | 0.825 | **0.659** | **0.480** | **0.443** | 0.599 |
| April | **0.676** | **0.682** | **0.520** | **0.639** | **0.445** | 0.691 | **0.580** | **0.524** | **0.599** | **0.535** | 0.778 | **0.557** | **0.677** | **0.488** | 0.602 |
| May | 0.796 | 0.714 | **0.648** | **0.572** | 0.627 | **0.481** | **0.245** | **0.637** | **0.626** | **0.203** | 0.701 | **0.170** | **0.607** | **0.211** | 0.544 |
| June | 0.754 | **0.494** | **0.592** | 0.790 | **0.478** | 0.614 | **0.373** | 0.324 | **0.655** | **0.311** | 0.844 | **0.536** | **0.527** | **0.620** | 0.583 |
| July | 0.599 | 0.335 | **0.480** | 0.450 | 0.201 | 0.681 | 0.161 | **0.520** | **0.673** | 0.173 | 0.778 | 0.095 | 0.235 | **0.265** | 0.414 |
| August | **0.669** | **0.390** | **0.572** | **0.571** | **0.251** | 0.571 | **0.286** | **0.633** | **0.638** | **0.192** | 0.811 | **0.429** | **0.367** | **0.444** | 0.506 |
| September | 0.732 | **0.590** | **0.659** | 0.740 | **0.514** | 0.682 | **0.409** | 0.715 | 0.725 | **0.304** | 0.802 | **0.152** | **0.403** | **0.410** | 0.591 |
| October | **0.646** | **0.345** | **0.563** | **0.476** | **0.303** | 0.841 | **0.594** | 0.809 | 0.760 | **0.674** | 0.891 | **0.703** | 0.703 | **0.669** | 0.650 |
| November | 0.723 | 0.766 | 0.864 | 0.803 | **0.575** | 0.912 | 0.801 | 0.764 | 0.811 | 0.853 | 0.856 | **0.633** | **0.594** | **0.096** | 0.725 |
| December | 0.711 | **0.692** | 0.915 | 0.797 | **0.500** | 0.886 | **0.656** | 0.718 | 0.770 | 0.701 | 0.909 | 0.782 | 0.822 | **0.046** | 0.730 |

### 4.2. Comparing Agri4Cast with PinovaMeteo

The second analysis (Agri4Cast vs. PinovaMeteo) includes the following parameters: air temperature and precipitation. Note that in this analysis, we use the data for 10 locations; one additional location (Dugi Otok) is added compared to the previous analysis since it is covered by the Agri4Cast dataset.

The correlation values presented in Table 6 compare air temperature data from the PinovaMeteo and Agri4Cast datasets and highlight several important observations:

- *Overall Reliability:* The data underscore a strong and consistently positive correlation between the two datasets, confirming their reliability for temperature monitoring.
- *Seasonal Variations:* While the overall correlation is robust, there is a slight decrease in correlation values during the winter months.
- *Inter-Station Differences:* Across most stations, the correlation values remain high. However, exceptions are observed at stations Potomje and Funtane, where the PinovaMeteo sensor records lower temperature values compared to Agri4Cast. This causes the temperature correlation values at these stations to fall below the threshold of 0.7 for several months.

The correlation values between the PinovaMeteo and Agri4Cast precipitation data reveal a varying relationship characterized by seasonal fluctuations and station-specific differences. Generally positive, the correlations are stronger during wetter months (e.g., May and June) and weaker during drier periods (e.g., November and December). Some stations exhibit stronger agreements (e.g., Našice and Otočac), while others show weaker correlations (e.g., Potomje and Suhopolje). Notably, negative correlations in certain instances, such as September 2020 at the Potomje station, indicate potential data discrepancies related to specific events. The lower correlation for some months may also be due to the fact that Agri4Cast provides data for a larger area (25 × 25 km), while in the event of localized or heavy rain, the amount of rainfall may vary across the segment.

**Table 6.** Correlations between PinovaMeteo and Agri4Cast data. Correlations below the threshold value of r = 0.7 are marked in bold. The month with the strongest correlation value for each location is highlighted in orange, the weakest in blue.

| Parameter | Year | Month | Našice | Kamenac | Suhopolje | Skenderovci | Nedelišče | Otočac | Oklaj | Potomje | Funtane | Dugi Otok |
|---|---|---|---|---|---|---|---|---|---|---|---|---|
| Temperature | 2019 | January | 0.842 | 0.954 | 0.938 | 0.906 | 0.970 | 0.918 | 0.946 | 0.846 | 0.716 | 0.859 |
| | | February | 0.946 | 0.962 | 0.946 | 0.891 | 0.973 | 0.883 | 0.917 | 0.723 | 0.761 | 0.892 |
| | | March | 0.928 | 0.908 | 0.939 | 0.913 | 0.959 | 0.873 | 0.927 | 0.827 | **0.660** | 0.786 |
| | | April | 0.956 | 0.922 | 0.949 | 0.943 | 0.979 | 0.954 | 0.975 | 0.937 | 0.904 | 0.941 |
| | | May | 0.969 | 0.965 | 0.977 | 0.984 | 0.983 | 0.958 | 0.922 | 0.842 | 0.922 | 0.896 |
| | | June | 0.963 | 0.969 | 0.942 | 0.936 | 0.928 | 0.964 | 0.983 | 0.913 | 0.949 | 0.925 |
| | | July | 0.940 | 0.905 | 0.901 | 0.932 | 0.929 | 0.940 | 0.952 | 0.790 | 0.911 | 0.858 |
| | | August | 0.946 | 0.902 | 0.898 | 0.934 | 0.908 | 0.940 | 0.973 | 0.792 | 0.911 | 0.868 |
| | | September | 0.968 | 0.937 | 0.980 | 0.949 | 0.975 | 0.962 | 0.971 | 0.840 | 0.963 | 0.912 |
| | | October | 0.985 | 0.971 | 0.963 | 0.930 | 0.955 | 0.821 | 0.907 | **0.525** | 0.932 | 0.907 |
| | | November | 0.949 | 0.962 | 0.945 | 0.864 | 0.974 | 0.907 | 0.884 | 0.796 | 0.876 | 0.859 |
| | | December | 0.938 | 0.980 | 0.967 | 0.830 | 0.982 | 0.941 | 0.968 | 0.879 | 0.883 | 0.944 |
| | 2020 | January | 0.907 | 0.953 | 0.935 | 0.738 | 0.924 | 0.830 | 0.798 | **0.531** | **0.695** | 0.840 |
| | | February | 0.840 | 0.920 | 0.901 | 0.830 | 0.886 | 0.916 | 0.896 | 0.739 | 0.792 | 0.857 |
| | | March | 0.970 | 0.966 | 0.981 | 0.970 | 0.983 | 0.976 | 0.957 | **0.693** | 0.906 | 0.954 |
| | | April | 0.954 | 0.970 | 0.978 | 0.985 | 0.986 | 0.945 | 0.943 | 0.862 | 0.877 | 0.888 |
| | | May | 0.847 | 0.907 | 0.915 | 0.919 | 0.923 | 0.873 | 0.926 | 0.921 | 0.898 | 0.842 |
| | | June | 0.917 | 0.937 | 0.943 | 0.962 | 0.952 | 0.929 | 0.942 | 0.831 | 0.919 | 0.927 |
| | | July | 0.940 | 0.923 | 0.943 | 0.969 | 0.913 | 0.963 | 0.929 | 0.860 | 0.806 | 0.797 |
| | | August | 0.941 | 0.867 | 0.943 | 0.919 | 0.929 | 0.930 | 0.942 | 0.681 | 0.906 | 0.796 |
| | | September | 0.974 | 0.939 | 0.974 | 0.932 | 0.973 | 0.961 | 0.984 | 0.889 | 0.944 | 0.967 |
| | | October | 0.979 | 0.974 | 0.961 | 0.837 | 0.981 | 0.922 | 0.959 | 0.860 | 0.880 | 0.934 |
| | | November | 0.981 | 0.989 | 0.976 | 0.962 | 0.981 | 0.876 | 0.886 | 0.728 | 0.894 | 0.951 |
| | | December | 0.979 | 0.990 | 0.968 | 0.864 | 0.978 | 0.907 | 0.858 | **0.656** | 0.868 | 0.858 |
| | 2021 | January | 0.946 | 0.987 | 0.967 | 0.961 | 0.985 | 0.958 | 0.963 | 0.931 | 0.905 | 0.945 |
| | | February | 0.978 | 0.982 | 0.984 | 0.961 | 0.976 | 0.972 | 0.982 | 0.907 | 0.902 | 0.968 |
| | | March | 0.944 | 0.929 | 0.957 | 0.918 | 0.950 | 0.909 | 0.897 | **0.663** | 0.850 | 0.907 |
| | | April | 0.974 | 0.971 | 0.976 | 0.975 | 0.983 | 0.946 | 0.959 | 0.901 | 0.871 | 0.922 |
| | | May | 0.901 | 0.899 | 0.932 | 0.940 | 0.901 | 0.853 | 0.944 | 0.875 | **0.601** | 0.877 |
| | | June | 0.896 | 0.879 | 0.920 | 0.948 | 0.921 | 0.800 | 0.923 | 0.851 | **0.597** | 0.853 |
| | | July | 0.920 | 0.930 | 0.956 | 0.952 | 0.952 | 0.961 | 0.864 | **0.626** | 0.764 | 0.751 |
| | | August | 0.984 | 0.974 | 0.987 | 0.978 | 0.966 | 0.967 | 0.987 | 0.850 | 0.903 | 0.936 |
| | | September | 0.967 | 0.938 | 0.973 | 0.958 | 0.935 | 0.933 | 0.955 | 0.901 | 0.793 | 0.847 |
| | | October | 0.976 | 0.977 | 0.978 | 0.945 | 0.972 | 0.952 | 0.955 | **0.687** | 0.871 | 0.941 |
| | | November | 0.897 | 0.917 | 0.924 | 0.873 | 0.948 | 0.747 | 0.883 | 0.780 | 0.752 | 0.904 |
| | | December | 0.942 | 0.977 | 0.947 | 0.902 | 0.918 | 0.889 | 0.922 | 0.910 | **0.599** | 0.856 |

**Table 6.** *Cont.*

| Parameter | Year | Month | Našice | Kamenac | Suhopolje | Skenderovci | Nedelišće | Otočac | Oklaj | Potomje | Funtane | Dugi Otok |
|---|---|---|---|---|---|---|---|---|---|---|---|---|
| | | | | | | | Stations | | | | | |
| Precipitation | 2019 | January | 0.232 | 0.283 | 0.400 | 0.222 | 0.652 | 0.565 | 0.897 | 0.771 | 0.592 | 0.939 |
| | | February | 0.820 | 0.897 | 0.746 | 0.905 | 0.399 | 0.215 | 0.715 | 0.821 | 0.990 | 0.516 |
| | | March | 0.559 | 0.168 | 0.420 | 0.389 | 0.903 | 0.420 | 0.402 | 0.461 | 0.786 | 0.914 |
| | | April | 0.674 | 0.821 | 0.724 | 0.439 | 0.676 | 0.488 | 0.410 | 0.813 | 0.268 | 0.194 |
| | | May | 0.892 | 0.616 | 0.934 | 0.518 | 0.855 | 0.836 | 0.573 | 0.962 | 0.059 | 0.753 |
| | | June | 0.519 | 0.753 | 0.686 | 0.525 | 0.838 | 0.840 | 0.574 | 0.931 | 0.913 | 0.669 |
| | | July | 0.650 | 0.537 | 0.740 | 0.328 | 0.670 | 0.841 | 0.880 | 0.473 | 0.646 | 0.732 |
| | | August | 0.190 | 0.637 | 0.273 | 0.085 | 0.753 | 0.611 | 0.766 | 0.128 | 0.596 | 0.978 |
| | | September | 0.371 | 0.559 | 0.670 | −0.070 | 0.728 | 0.813 | 0.124 | 0.382 | −0.045 | 0.850 |
| | | October | 0.856 | 0.957 | 0.784 | 0.282 | 0.905 | 0.861 | 0.980 | 0.975 | 0.169 | 0.815 |
| | | November | 0.432 | 0.392 | 0.715 | 0.543 | 0.646 | 0.317 | 0.403 | 0.749 | 0.160 | 0.507 |
| | | December | 0.759 | 0.779 | 0.682 | 0.784 | 0.852 | 0.948 | 0.745 | 0.566 | 0.846 | 0.839 |
| | 2020 | January | 0.967 | 0.958 | 0.982 | 0.974 | 0.995 | 0.947 | 0.567 | 0.961 | 0.788 | 0.853 |
| | | February | 0.852 | 0.661 | 0.899 | 0.868 | 0.976 | 0.386 | 0.742 | 0.500 | −0.028 | 0.398 |
| | | March | 0.223 | 0.376 | 0.473 | 0.134 | 0.693 | 0.310 | 0.897 | 0.609 | 0.914 | 0.903 |
| | | April | 0.789 | 0.701 | 0.627 | 0.866 | 0.816 | 0.544 | 0.809 | 0.868 | 0.303 | 0.636 |
| | | May | 0.444 | 0.184 | 0.197 | 0.628 | 0.143 | 0.383 | 0.297 | 0.117 | 0.119 | 0.414 |
| | | June | 0.297 | 0.372 | 0.473 | 0.580 | 0.206 | 0.604 | 0.612 | 0.655 | 0.587 | 0.753 |
| | | July | 0.642 | −0.099 | 0.613 | 0.059 | 0.809 | 0.534 | 0.089 | 0.142 | 0.045 | 0.994 |
| | | August | 0.955 | 0.957 | 0.531 | 0.430 | 0.683 | 0.873 | 0.542 | 0.999 | | 0.392 |
| | | September | 0.828 | 0.615 | 0.883 | 0.798 | 0.476 | 0.856 | 0.879 | 0.619 | | 0.764 |
| | | October | 0.672 | 0.691 | 0.565 | 0.570 | 0.811 | 0.879 | 0.631 | 0.141 | | 0.756 |
| | | November | 0.475 | 0.704 | 0.302 | 0.565 | 0.964 | −0.048 | 0.774 | 0.412 | −0.103 | 0.933 |
| | | December | 0.743 | 0.699 | 0.902 | 0.670 | 0.854 | 0.537 | 0.819 | 0.888 | −0.159 | 0.619 |
| | 2021 | January | 0.708 | 0.898 | 0.900 | 0.762 | 0.846 | 0.481 | 0.890 | | −0.233 | 0.582 |
| | | February | 0.650 | 0.876 | 0.867 | 0.549 | 0.715 | 0.715 | 0.850 | | −0.131 | 0.463 |
| | | March | 0.860 | 0.955 | 0.990 | 0.969 | 0.974 | 0.013 | 0.766 | | −0.140 | 0.949 |
| | | April | 0.432 | 0.513 | 0.479 | 0.304 | 0.696 | 0.240 | 0.444 | | 0.067 | 0.430 |
| | | May | 0.698 | 0.772 | 0.431 | 0.628 | 0.675 | 0.814 | 0.616 | | −0.010 | −0.010 |
| | | June | 0.658 | 0.760 | −0.074 | 0.958 | 0.906 | 0.479 | 0.726 | | 0.271 | 0.252 |
| | | July | 0.572 | 0.610 | 0.964 | 0.467 | 0.413 | 0.021 | 0.866 | | 0.715 | 0.226 |
| | | August | 0.138 | −0.030 | 0.723 | 0.367 | 0.911 | 0.251 | 0.587 | | 0.754 | 0.045 |
| | | September | 0.433 | | 0.571 | 0.499 | 0.594 | 0.870 | 0.823 | | 0.366 | 0.895 |
| | | October | 0.823 | 0.861 | 0.624 | 0.864 | 0.808 | 0.672 | 0.906 | | 0.774 | 0.575 |
| | | November | 0.571 | 0.716 | 0.631 | 0.832 | 0.421 | 0.558 | 0.616 | | 0.772 | 0.483 |
| | | December | 0.721 | 0.686 | 0.864 | 0.857 | 0.813 | 0.854 | 0.681 | | 0.861 | 0.643 |

### 4.3. Comparing ERA5-Land with Agri4Cast

For additional examination of remote sensing sources, we conducted a correlation analysis involving the ERA5-Land and Agri4Cast datasets, with the inclusion of the additional parameters listed in Tables A1 and A2. The results are included in the Appendix A (Tables A3–A5). Temperature consistently displays strong positive correlations across months and stations. Precipitation correlations vary, with some locations showing high positive correlations during specific months, while others have low or negative correlations, likely due to regional weather patterns and local climate influences. Evapotranspiration correlations fluctuate across months, mainly falling within the moderate to high range and are influenced by factors like temperature and humidity. While evapotranspiration generally correlates positively with temperature, discrepancies, especially in late summer, may be attributed to differences between the Copernicus and Agri4Cast datasets. Radiation correlations are mostly positive, implying seasonal solar radiation fluctuations, and align

with temperature. Notably, wind speed correlations remain strong across months and stations, with some locations showing exceptionally robust correlations during specific months. The positive correlation between wind speed and temperature is due to its role in heat transport, and coastal areas maintain high correlations even in summer due to prevailing strong winds.

### 4.4. Significance Test

To determine the significance of differences in the variables between the data sources, the *t*-test was employed. Parametric statistical tests, such as the *t*-test, are statistical methods that assume that the data come from the same Gaussian distribution, i.e., a distribution of data with the same mean and standard deviation. The test provides a *p*-value that can be used to interpret the test result. The *p*-value can be viewed as the probability of observing the two data samples under the basic assumption (null hypothesis) that the two samples were drawn from a population with the same distribution. The *p*-value can be interpreted in the context of a chosen significance level, called alpha. The chosen value for $\alpha$ is 5% or 0.05. If the *p*-value is below the significance level, the test says that there is sufficient evidence to reject the null hypothesis and that the samples were probably drawn from populations with different distributions.

Mathematically, the *t*-test involves taking a sample from each of the two sets and framing the problem by assuming a null hypothesis that posits equality of the two means. Subsequently, certain values are computed based on relevant formulas and compared to standard values, leading to the acceptance or rejection of the null hypothesis as appropriate. The key assumptions in this context are the following:

- Observations in each sample are independent and identically distributed,
- Observations in each sample follow a normal distribution,
- Observations in each sample possess equal variances.

In this study, *t*-tests were conducted to determine whether the data collected from the ground-based sensors, PinovaMeteo stations, have a significant difference from the satellite data sourced from Copernicus CDS and the Agri4Cast portal.

We conducted a comprehensive analysis of Copernicus and PinovaMeteo data gathered from nine stations, assessing the statistical significance for each month over the course of three years. Figure 2a visually presents this analysis in relation to the air temperature parameter. One of the notable observations arising from our analysis is the consistent divergence of location Potomje from the established confidence interval. Additionally, we identified deviations from the confidence interval in the data points of Funtane, Otočac, and Oklaj. In contrast, the remaining stations within the dataset demonstrate statistical significance, indicating a strong alignment between the Copernicus data and in situ measurements.

Regarding the less favorable performance of the *t*-test with precipitation data, we observed consistent statistical insignificance in station comparisons.

As part of our research, we conducted a *t*-test involving 10 stations, comparing Agri4Cast and PinovaMeteo data with the results for air temperature illustrated in Figure 2b. Notably, Potomje consistently displayed statistical insignificance across all months spanning three years. Furthermore, the comparison of Pinova and Agri4Cast data revealed that the same stations consistently fell into the non-significant category, including Otočac, Funtane, and Oklaj. This pattern suggests the presence of specific *"problematic"* locations within the dataset.

However, when examining the data related to precipitation, we noticed a further reduction in the number of statistically significant stations compared to the findings obtained from the Copernicus data analysis. This decrease in significance may be attributed to various factors, including measurement discrepancies, localized variations, or potential limitations in accurately representing precipitation through satellite data in specific regions. To gain a deeper understanding and establish more precise models for precipitation in these areas, further investigation and potential enhancements in data sources are required.

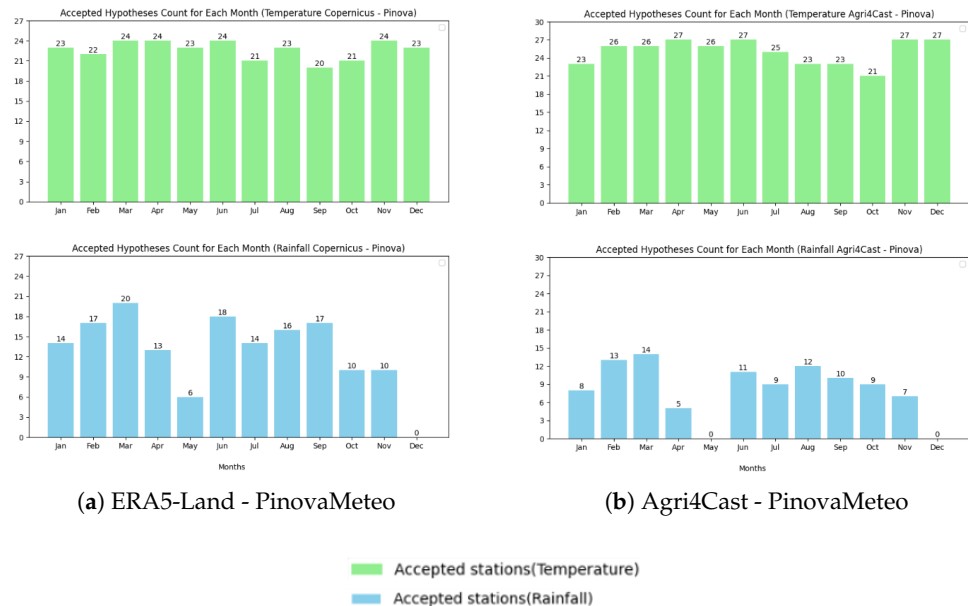

(**a**) ERA5-Land - PinovaMeteo        (**b**) Agri4Cast - PinovaMeteo

**Figure 2.** *t*-test analysis of air temperature and precipitation data.

### 4.5. Principal Component Analysis

The statistical analysis was carried out using the ad hoc Python functions, based on the scipy and sklearn libraries. The focus of the analysis was only on the symmetric data, which were available for the variables air temperature and precipitation from all three data sources. Prior to the analysis, the data were detrended by subtracting each value from the variable mean. Annual cycles were also removed from the data by calculating the mean value for each month at each location, and then subtracting this mean value from the corresponding data points. This process removed all annual cycles present in the data, resulting in a dataset that was both detrended and devoid of annual cycles. This refined dataset was then used for further statistical analysis. The variables of interest, air temperature and precipitation, were now prepared in a manner that allowed for a more accurate and focused comparison between the different data sources.

The next step in our analysis was to calculate the residuals, considering PinovaMeteo (terrestrial data source) as our ground truth and subtracting values from ERA5-Land and Agri4Cast datasets at monthly mean basis. This resulted in four derived variables:

- PinovaMeteo-ERA5-Land_prec,
- PinovaMeteo-agri_prec,
- PinovaMeteo-ERA5-Land_temp,
- PinovaMeteo-agri_temp.

These variables represent the daily and monthly mean differences in precipitation and air temperature readings between PinovaMeteo and the other two sources for each observed location.

In our analysis, principal component analysis (PCA) was applied to both the detrended and de-annualized monthly means and the residuals. The purpose of applying PCA to these datasets is to reduce their dimensionality and identify the key variables responsible for most of the variability in the data. The aim of applying PCA to the residuals is to identify patterns or structures in the residuals that may not be apparent when looking at the residuals alone and to emphasize systematic errors.

Results of PCA

Analysis of the residuals between different data sources shows considerable differences in the forecast accuracy for both precipitation and air temperature on a daily (Figure 3) and monthly basis (Figure 4). The daily precipitation data are zero-inflated,

resulting in residuals tending towards zero on most days (as shown in the top two graphs of Appendix B Figure A1). However, on days with predicted precipitation, ERA5-Land reveals more positive deviations from the ground-based measurements (the upper left graph in Figure 3), but these deviations are smaller in magnitude. In contrast, the Agri4Cast data source shows predominantly negative daily deviations from the ground-based measurements (the upper right graph in Figure 3), followed by a larger standard deviation of the residuals. For air temperature data, both ERA5-Land and Agri4Cast show similar patterns of deviations and magnitudes compared to the ground-based stations (PinovaMeteo), as can be observed in the two lower graphs in Figures 3 and A1. It can be observed that the residuals are consistently the smallest for location Suhopolje. A *t*-test performed between locations with complex or plain topography did not reveal significant differences between group means. However, daily deviations observed during the agricultural growing season can in practice have a large economic impact on crop production.

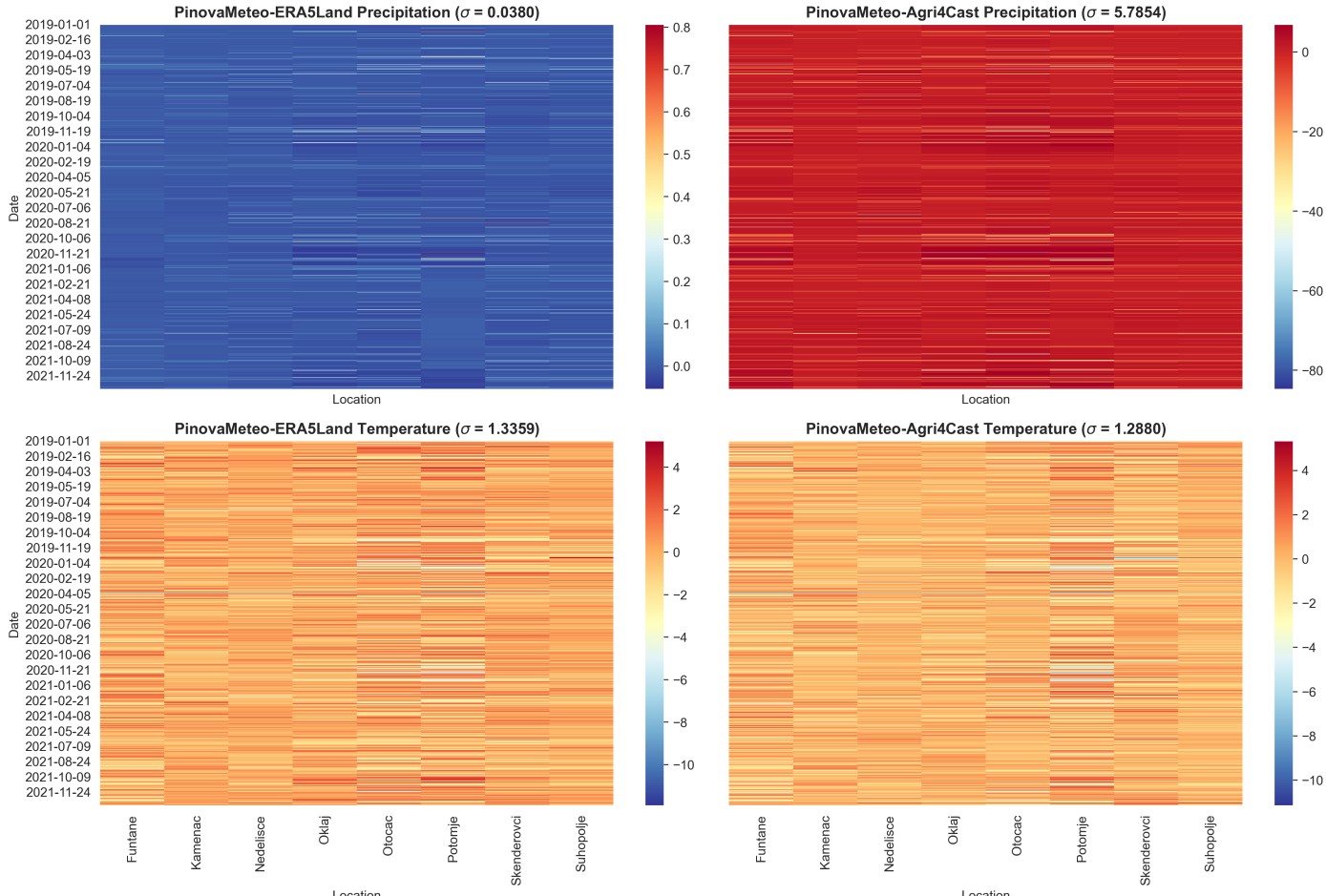

**Figure 3.** Heatmap of residuals of daily mean air temperature and precipitation between data source PinovaMeteo (ground-based) and two model-based data sources ERA5-Land and Agri4Cast. Standard deviations are shown in plot main titles.

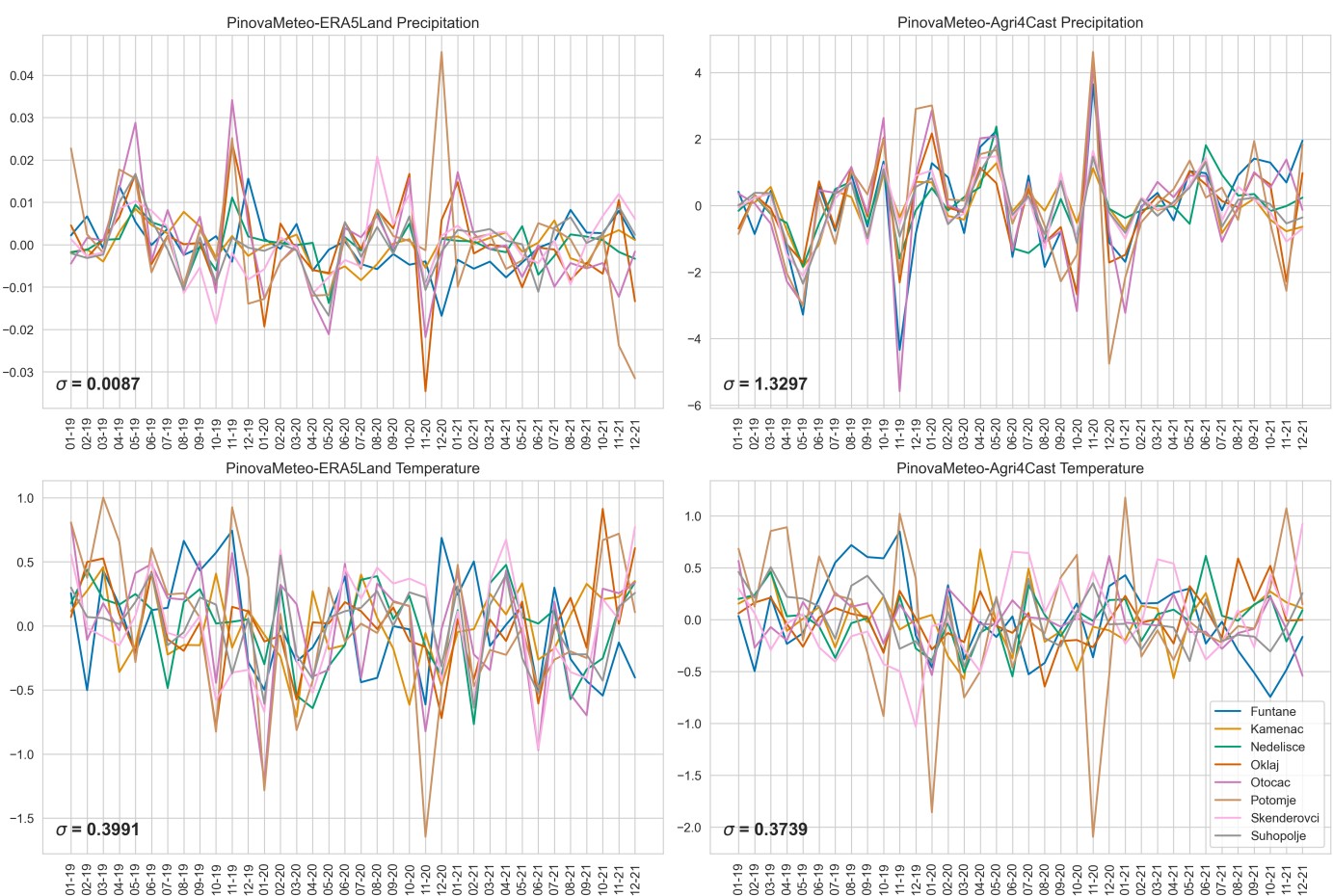

**Figure 4.** Residuals of monthly mean temperature and precipitation between data source PinovaMeteo (ground-based) and two model-based data sources Era5-Land and Agri4Cast. Standard deviations are shown within plots in bold.

The monthly residual values follow Gaussian properties due to sampling. The highest monthly mean values of residuals between PinovaMeteo and ERA5-Land for precipitation were observed for the locations Potomje, Oklaj, and Otočac, for the months of November in 2019 and 2020, and December in 2020 (location Potomje), as can be seen in the upper left graph in Figure 4. Similarly, the highest monthly mean values of the residuals between PinovaMeteo and Agri4Cast for precipitation were observed for the months of November in 2019 and 2020, with the highest leverage of locations Otočac and Funtane in 2019, and Potomje, Oklaj, Otočac and Funtane in 2020 (the upper right graph in Figure 4). Note also that the magnitude of the residual values is significantly higher in the case of PinovaMeteo-Agri4Cast compared to PinovaMeteo-ERA5-Land for precipitation. Analysis of the detrended residuals of air temperature also indicates discrepancies between the assessed data sources, with monthly residual mean values often exceeding 0.5 °C, and sometimes even 1 °C at some locations (the two lower graphs in Figures 4 and A2), while both ERA5-Land and Agri4Cast show similar patterns of deviations. Note that the observed temperature differences may seem small, but are very important for PA as they can have a significant impact on crop health and agricultural production in general.

PCA analysis of the monthly mean data, as illustrated in Figure 5, did not show support for grouping between locations. The first component was mostly negatively correlated to temperature data from all analyzed data sources, while the second component was mostly positively correlated with precipitation data from all data sources. The first two components explained 83.1% of the overall variance.

PCA analysis of the residuals (Figure 6) showed the highest absolute loading weights (eigenvectors) for PinovaMeteo-Agri4Cast for precipitation and PinovaMeteo-ERA5-Land for temperature, showing the highest absolute values of the residuals for these pairs. Despite the lack of a distinct grouping of locations, the information about high scattering of the residuals in locations Potomje and Otočac indicates clear discrepancies and possible systematic errors between different data sources at these locations (Figure 5). The first two components explained 85.8% of the overall variance.

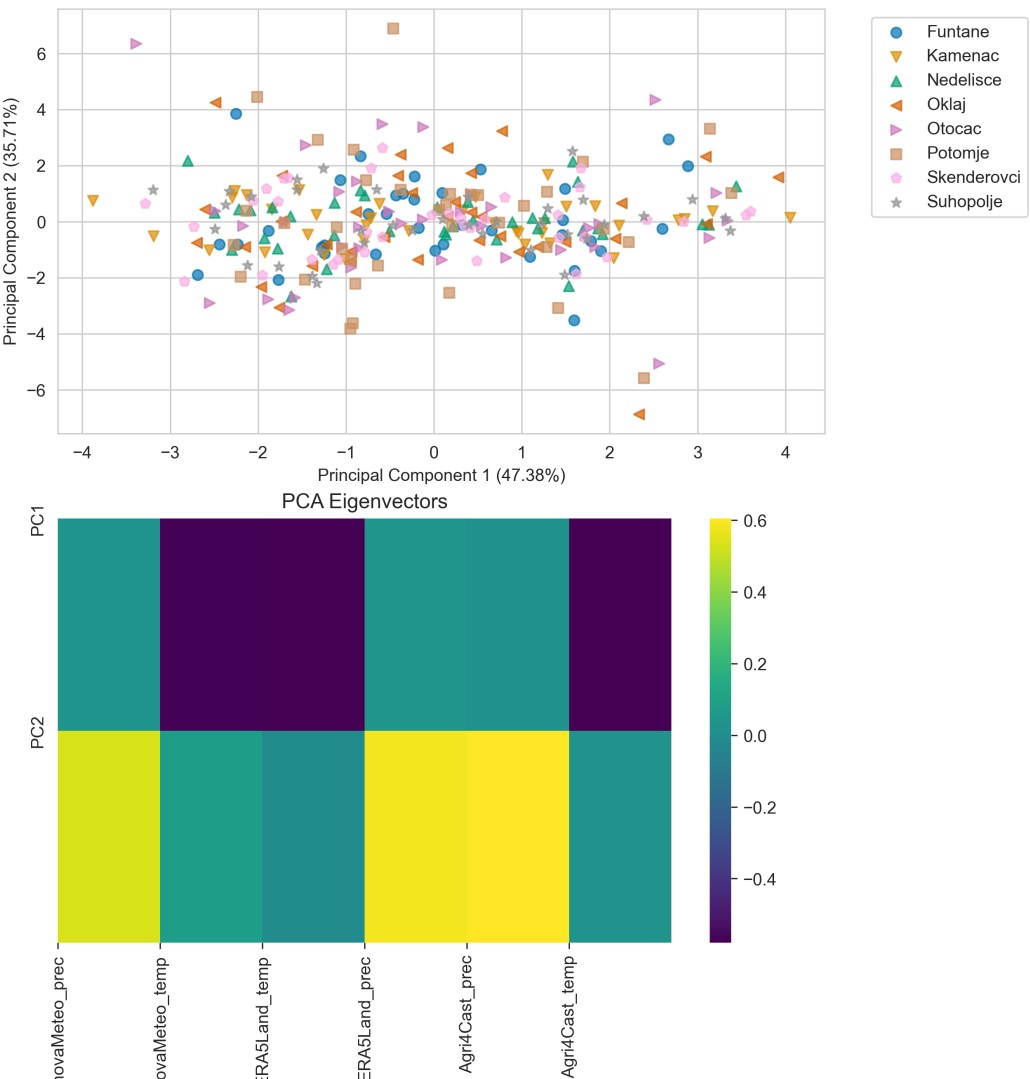

**Figure 5.** PCA of the monthly mean values across 8 locations covered by this study (**above**) and eigenvector weights (correlations with PCs) shown as heatmap (**below**).

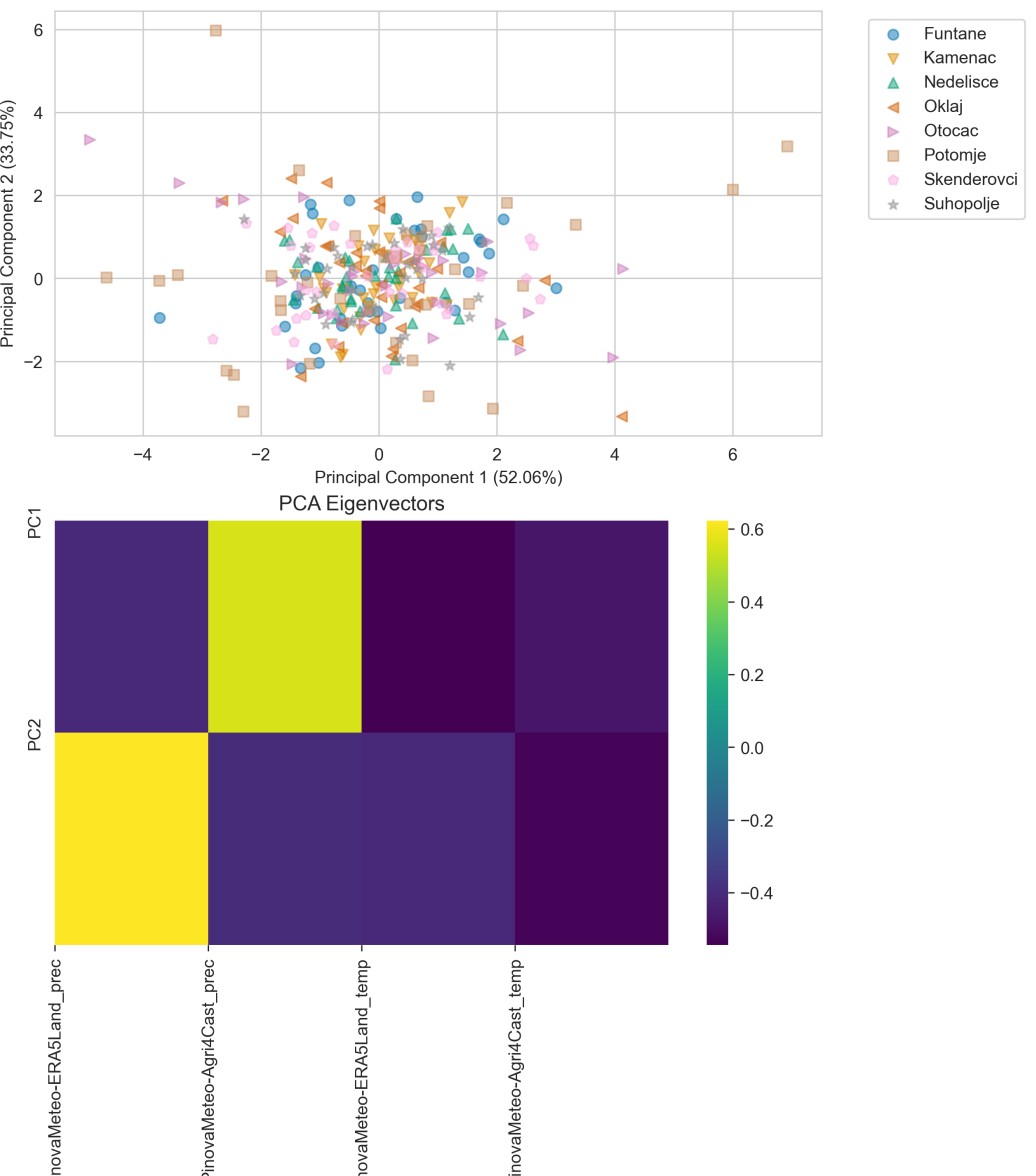

**Figure 6.** PCA of the monthly residuals between PinovaMeteo (ground-based) and two model-based data sources (ERA5-Land and Agri4Cast) across 8 locations covered by this study (**above**) and eigenvector weights (correlations with PCs) shown as heatmap (**below**).

## 5. Discussion

In this paper, we compare data from sources that provide freely available meteorological data that can be used in precision agriculture with data from on-site agrometeorological stations. Specifically, we analyze and compare data types from the Copernicus CDS and the Agri4Cast meteorological database with measurements from the PinovaMeteo stations deployed at 10 sites in Croatia for a three-year period. The analyzed agrometeorological parameters include air and soil temperature, and precipitation.

Examining the air temperature correlations, we find strong positive correlations, especially between the data collected by the PinovaMeteo stations and the ERA5-Land dataset. This could be due to the smaller grid cells in the Copernicus dataset compared to Agri4Cast. In general, the temperature correlations are lower in the winter months. A notable observation is that the PinovaMeteo stations tend to record more extreme temperatures, while the ERA5-Land and Agri4Cast datasets show greater stability. For example, the PinovaMeteo data consistently show lower temperatures, especially in winter, compared to the other two sources. In contrast, Agri4Cast generally records higher temperatures.

However, the precipitation correlations are, in general, moderate, or even weak or negative for specific months. Regarding precipitation data, the PinovaMeteo stations detect precipitation events in a timely manner, while Agri4Cast and Copernicus CDS often register them with some delay. The correlation results between the PinovaMeteo and Agri4Cast data generally show stronger correlations compared to PinovaMeteo and ERA5-Land. However, the overall correlation weakens in the summer months, which are characterized by sporadic and sudden precipitation.

Correlations between soil temperatures were calculated for five sites using Copernicus and PinovaMeteo data. In particular, the Našice location show consistently high and relatively linear correlation trends, with somewhat lower values in the summer months.

An analysis of the residuals indicated higher overall residuals for precipitation between PinovaMeteo and Agri4Cast than PinovaMeteo and ERA5-Land, while the residuals for air temperature were comparable. The precipitation accuracy in this case might be affected by better resolution of the ERA5-Land dataset compared to Agri4Cast.

The principal component analysis indicated potential systematic errors of both hourly estimates from ERA5-Land and daily estimates from Agri4Cast for locations near the sea or large water bodies (location Potomje) and with complex topography (location Otočac). This is consistent with the existing reports that have identified topographic properties to negatively influence the accuracy of satellite readings, especially in complex terrains [30].

Deviations of residuals on daily and monthly scales found in our study show limitations of satellite data for decision-making in agriculture, especially considering that sometimes thresholds for taking measures are below 1 °C (frost protection, ET-based irrigation).

## 6. Conclusions

Our study uses three-year agrometeorological data retrieved for 10 different locations in Croatia from three data sources relevant for agriculture: ground-based weather stations (PinovaMeteo), gridded agrometeorological data with $25 \times 25$ km resolution (Agri4Cast), and gridded meteorological data with $9 \times 9$ km resolution (ERA5-Land). The findings of our study have several important implications for data-driven agriculture and can be summarized as follows:

- The study confirms high correlations between remote sensing data and ground observations and can be used to monitor air temperature with high accuracy, even in locations with complex topography.
- The study highlights the limitations of remote sensing for monitoring precipitation. The study found that precipitation data from different sources showed very low resemblance to ground observations, regardless of topography. This suggests that remote sensing cannot be relied upon to provide accurate precipitation estimates.
- The study indicates that remote sensing may introduce larger and systematic errors for locations near the sea or large water bodies and with complex topography. This is likely due to the fact that satellite data are affected by factors such as cloud cover, atmospheric aerosols, and land surface conditions.
- The study found that deviations of residuals on daily and monthly scales can be significant in use cases of PA, even for air temperature. This means that satellite data may not be accurate enough to support decision-making in agriculture, especially considering that thresholds for taking measures are sometimes below 1 °C (frost protection, ET-based irrigation).

Overall, the study's findings suggest that PA requires ground-based measurements, especially in areas with complex topography and near large water bodies or the sea. Although our data did not reveal significant differences between data source residuals based on topography, the observed daily deviations, especially within the agricultural growing season, can create a large impact on crop growth. A significantly larger dataset with ground observations from a larger number of representative stations located in geographical regions with similar topology is needed for further study.

Future research should, thus, focus on the following: (1) the development of new and improved methods for correcting systematic errors in remote sensing data, especially for precipitation and in areas with complex topography and near large water bodies or the sea; (2) fusing of remote sensing data with other sources of data, such as ground-based measurements and numerical models; and (3) development of new and improved decision support systems that can account for the uncertainty in remote sensing data. With such improvements, the use of fused data from remote sensing and ground-based sources can provide reliable information which is effective for PA.

**Author Contributions:** Each author made substantial contributions to this publication. Conceptualization, I.P.Ž., M.K. and V.G.; methodology, D.K. and V.G.; formal analysis, V.G. and D.K.; writing—original draft preparation, D.K., I.P.Ž., V.G. and K.T.; writing—review and editing, I.P.Ž., D.K. and V.G.; supervision, I.P.Ž. All authors have read and agreed to the published version of the manuscript.

**Funding:** This work has been supported in part by the project IoT-field: *An Ecosystem of Networked Devices and Services for IoT Solutions Applied in Agriculture* funded by the European Union from the European Regional Development Fund and by the Croatian Science Foundation under the project IP-2019-04-1986 IoT4us: *Smart human-centric services in interoperable and decentralised IoT environments*.

**Data Availability Statement:** The data that support the findings of this study are available by request from the corresponding author.

**Acknowledgments:** This work has been supported in part by the project IoT-field: *An Ecosystem of Networked Devices and Services for IoT Solutions Applied in Agriculture* funded by the European Union from the European Regional Development Fund and by the Croatian Science Foundation under the project IP-2019-04-1986 IoT4us: *Smart human-centric services in interoperable and decentralised IoT environments*. We would like to express our sincere gratitude to Pinova for the permission to use the selected readings from PinovaMeteo stations in this study.

**Conflicts of Interest:** The authors declare no conflicts of interest. The funders had no role in the design of the study; in the collection, analyses, or interpretation of data; in the writing of the manuscript, or in the decision to publish the results.

## Appendix A. ERA5-Land—Agri4Cast Comparison

Tables A3–A5 show the monthly correlation values between the ERA5-Land and Agri4Cast datasets. The parameters that were chosen for comparison are available in both datasets and include the following: air temperature, precipitation, evapotranspiration, radiation, and wind speed. The descriptions of the additional parameters are provided in Tables A1 and A2.

**Table A1.** Additional ERA5-Land parameters for correlation analysis.

| Name | Unit | Description |
|---|---|---|
| 10 m u-component of wind | m/s | eastward component of the 10 m wind |
| 10 m v-component of wind | m/s | northward component of the 10 m wind |
| surface pressure | Pa | atmosphere pressure of the on the surface of land, sea, and inland water |
| surface net solar radiation | $J/m^2$ | amount of solar radiation |
| total evaporation | m of water equivalent | accumulated amount of water that has evaporated from the Earth's surface |

**Table A2.** Additional Agri4Cast parameters for correlation analysis.

| Name | Unit | Description |
|---|---|---|
| wind speed | m/s | average wind speed at 10 m altitude |
| pressure | hPa | vapour pressure |
| radiation | $kJ/m^2$ | solar radiation per day |
| evaporation | mm | potential evapotranspiration per day |

**Table A3.** 2019 Correlations between ERA5-Land and Agri4Cast data. Correlations below the correlation threshold value of r = 0.7 are marked in bold. The month with the strongest correlation value for each location is highlighted in orange, the weakest in blue.

| Station | Parameter | Jan | Feb | Mar | Apr | May | Jun | Jul | Aug | Sep | Oct | Nov | Dec |
|---|---|---|---|---|---|---|---|---|---|---|---|---|---|
| Našice | Temperature | 0.882 | 0.952 | 0.914 | 0.945 | 0.964 | 0.944 | 0.942 | 0.965 | 0.946 | 0.982 | 0.928 | 0.889 |
| | Precipitation | 0.852 | 0.939 | **0.263** | **0.553** | 0.786 | **0.563** | **0.672** | **0.286** | **0.411** | **0.652** | **0.161** | **0.624** |
| | Evapotranspiration | 0.780 | 0.798 | **0.571** | 0.893 | 0.862 | 0.927 | 0.798 | 0.708 | **0.604** | **0.606** | 0.887 | **0.520** |
| | Radiation | 0.790 | 0.954 | 0.915 | 0.930 | 0.945 | 0.929 | 0.871 | 0.754 | 0.907 | 0.878 | 0.741 | 0.798 |
| | Wind speed | 0.882 | 0.952 | 0.914 | 0.945 | 0.964 | 0.944 | 0.942 | 0.965 | 0.946 | 0.982 | 0.928 | 0.889 |
| Kamenac | Temperature | 0.921 | 0.928 | 0.924 | 0.957 | 0.962 | 0.955 | 0.948 | 0.956 | 0.934 | 0.977 | 0.933 | 0.890 |
| | Precipitation | 0.794 | 0.960 | **0.253** | **0.576** | 0.785 | 0.727 | 0.770 | **0.284** | **0.659** | 0.661 | **0.233** | 0.772 |
| | Evapotranspiration | 0.886 | 0.816 | **0.641** | 0.848 | 0.880 | 0.901 | **0.644** | **0.053** | **0.059** | **0.604** | 0.818 | **0.576** |
| | Radiation | 0.753 | 0.924 | 0.881 | 0.907 | 0.950 | 0.904 | 0.871 | 0.804 | 0.910 | 0.869 | **0.639** | 0.761 |
| | Wind speed | 0.958 | 0.978 | 0.888 | 0.893 | 0.926 | 0.872 | 0.834 | 0.842 | 0.820 | 0.926 | 0.925 | 0.951 |
| Suhopolje | Temperature | 0.921 | 0.957 | 0.908 | 0.943 | 0.963 | 0.926 | 0.920 | 0.944 | 0.918 | 0.973 | 0.920 | 0.943 |
| | Precipitation | **0.679** | 0.757 | **0.413** | **0.596** | 0.872 | **0.373** | **0.618** | **0.206** | **0.328** | 0.724 | **0.316** | **0.648** |
| | Evapotranspiration | 0.852 | 0.779 | 0.703 | 0.897 | 0.853 | 0.883 | 0.865 | **0.698** | **0.679** | 0.722 | 0.851 | **0.422** |
| | Radiation | 0.727 | 0.909 | 0.906 | 0.922 | 0.948 | 0.921 | 0.909 | 0.703 | 0.905 | 0.830 | 0.731 | 0.795 |
| | Wind speed | 0.906 | 0.918 | 0.898 | 0.747 | 0.907 | 0.784 | 0.777 | 0.844 | 0.713 | 0.887 | **0.606** | 0.922 |
| Skenderovci | Temperature | 0.913 | 0.959 | 0.902 | 0.963 | 0.978 | 0.951 | 0.954 | 0.960 | 0.963 | 0.969 | 0.893 | 0.927 |
| | Precipitation | 0.814 | 0.951 | **0.352** | **0.560** | 0.763 | **0.606** | **0.485** | **0.382** | **0.458** | **0.536** | **0.197** | 0.778 |
| | Evapotranspiration | **0.401** | **0.573** | **0.482** | 0.906 | 0.879 | 0.932 | 0.869 | 0.817 | 0.799 | **0.597** | 0.783 | **0.068** |
| | Radiation | 0.723 | 0.967 | 0.952 | 0.934 | 0.905 | 0.930 | 0.840 | **0.674** | 0.887 | 0.874 | 0.765 | 0.835 |
| | Wind speed | 0.837 | 0.934 | 0.888 | 0.811 | 0.909 | **0.661** | 0.753 | 0.904 | 0.755 | 0.791 | **0.419** | 0.765 |
| Nedelišće | Temperature | 0.908 | 0.933 | 0.933 | 0.953 | 0.948 | 0.905 | 0.923 | 0.858 | 0.891 | 0.954 | 0.942 | 0.977 |
| | Precipitation | **0.613** | **0.236** | 0.835 | 0.711 | 0.849 | **0.667** | **0.504** | 0.770 | **0.190** | 0.654 | **0.489** | 0.918 |
| | Evapotranspiration | **0.294** | 0.807 | **0.333** | 0.812 | **0.593** | 0.820 | 0.824 | 0.834 | 0.724 | **0.613** | 0.752 | **0.265** |
| | Radiation | 0.581 | 0.845 | 0.926 | 0.946 | 0.936 | 0.928 | 0.869 | 0.852 | 0.907 | 0.865 | 0.743 | 0.744 |
| | Wind speed | 0.861 | 0.948 | 0.825 | 0.856 | 0.858 | 0.830 | **0.603** | 0.863 | 0.768 | 0.884 | 0.839 | 0.925 |
| Funtane | Temperature | 0.687 | 0.743 | 0.768 | 0.895 | 0.855 | 0.967 | 0.880 | 0.874 | 0.898 | 0.837 | 0.771 | 0.904 |
| | Precipitation | 0.700 | 0.973 | **0.313** | **0.263** | **0.609** | **0.496** | 0.941 | 0.810 | **0.511** | 0.857 | **0.680** | 0.879 |
| | Evapotranspiration | 0.744 | **0.652** | 0.857 | **0.411** | **0.115** | 0.808 | **−0.124** | **0.372** | **0.687** | 0.822 | **0.271** | 0.714 |
| | Radiation | 0.840 | 0.965 | 0.930 | 0.911 | 0.884 | 0.812 | 0.894 | 0.833 | 0.943 | 0.825 | 0.923 | 0.943 |
| | Wind speed | 0.706 | 0.852 | 0.863 | **0.696** | 0.838 | **0.671** | 0.838 | 0.852 | 0.792 | 0.748 | 0.917 | 0.790 |
| Otočac | Temperature | 0.788 | 0.915 | 0.756 | 0.870 | 0.898 | 0.864 | 0.841 | 0.754 | 0.841 | 0.772 | 0.810 | 0.874 |
| | Precipitation | **0.665** | **0.475** | **0.686** | **0.481** | 0.736 | 0.871 | 0.865 | **0.199** | **0.337** | **0.637** | **0.226** | 0.961 |
| | Evapotranspiration | 0.045 | **0.424** | **0.637** | 0.856 | 0.837 | 0.847 | 0.743 | 0.787 | **0.643** | 0.360 | **0.506** | **0.456** |
| | Radiation | **0.652** | 0.879 | 0.902 | 0.870 | 0.874 | 0.807 | 0.875 | 0.864 | 0.773 | 0.748 | 0.588 | 0.691 |
| | Wind speed | 0.794 | 0.940 | 0.711 | 0.724 | 0.832 | 0.768 | 0.520 | 0.834 | 0.615 | 0.806 | 0.604 | 0.763 |
| Oklaj | Temperature | 0.872 | 0.861 | 0.853 | 0.945 | 0.915 | 0.951 | 0.885 | 0.888 | 0.877 | 0.858 | 0.881 | 0.932 |
| | Precipitation | 0.797 | 0.783 | **0.590** | **0.525** | **0.380** | 0.930 | **0.235** | **0.078** | 0.498 | 0.871 | **0.449** | 0.811 |
| | Evapotranspiration | **0.457** | **0.606** | 0.814 | 0.832 | 0.837 | 0.756 | **0.363** | **0.220** | 0.443 | 0.735 | **0.503** | **0.642** |
| | Radiation | 0.806 | 0.937 | 0.808 | 0.887 | 0.884 | 0.898 | 0.835 | 0.756 | 0.878 | 0.818 | 0.843 | 0.906 |
| | Wind speed | 0.885 | 0.929 | 0.912 | 0.748 | 0.878 | 0.837 | 0.744 | 0.832 | 0.906 | 0.891 | 0.851 | 0.908 |
| Potomje | Temperature | 0.925 | 0.897 | 0.952 | 0.918 | 0.939 | 0.977 | 0.935 | 0.907 | 0.946 | 0.924 | 0.856 | 0.912 |
| | Precipitation | 0.739 | 0.851 | **0.250** | **0.676** | 0.856 | | 0.938 | **0.502** | **0.224** | **0.483** | **0.539** | **0.544** |
| | Evapotranspiration | **0.074** | **0.318** | 0.778 | **0.599** | **0.377** | **0.505** | **0.029** | **0.356** | **0.317** | 0.764 | **0.550** | **0.285** |
| | Radiation | 0.811 | 0.950 | 0.839 | 0.818 | 0.889 | 0.874 | 0.844 | 0.756 | 0.833 | 0.852 | 0.917 | 0.887 |
| | Wind speed | 0.708 | 0.871 | 0.801 | 0.902 | 0.767 | 0.796 | 0.886 | 0.867 | 0.770 | 0.736 | 0.902 | 0.759 |

**Table A4.** 2020 Correlations between ERA5-Land and Agri4Cast data. Correlations below the correlation threshold value of r = 0.7 are marked in bold. The month with the strongest correlation value for each location is highlighted in orange, the weakest in blue.

| Station | Parameter | Jan | Feb | Mar | Apr | May | Jun | Jul | Aug | Sep | Oct | Nov | Dec |
|---|---|---|---|---|---|---|---|---|---|---|---|---|---|
| Našice | Temperature | 0.908 | 0.916 | 0.979 | 0.966 | 0.889 | 0.895 | 0.910 | 0.862 | 0.955 | 0.974 | 0.968 | 0.960 |
| | Precipitation | 0.811 | **0.675** | 0.715 | 0.782 | **0.448** | 0.321 | 0.862 | **0.463** | 0.659 | 0.535 | **0.674** | **0.588** |
| | Evapotranspiration | 0.842 | **0.461** | 0.765 | 0.856 | **0.659** | 0.867 | **0.690** | 0.602 | **0.432** | 0.618 | 0.719 | **0.531** |
| | Radiation | 0.782 | 0.885 | 0.891 | 0.903 | 0.840 | 0.922 | 0.882 | 0.955 | 0.943 | 0.950 | 0.817 | 0.828 |
| | Wind speed | 0.912 | 0.966 | 0.937 | 0.930 | 0.915 | 0.974 | 0.834 | 0.728 | 0.912 | 0.927 | 0.921 | 0.921 |
| Kamenac | Temperature | 0.877 | 0.951 | 0.982 | 0.975 | 0.920 | 0.923 | 0.897 | 0.779 | 0.947 | 0.972 | 0.964 | 0.953 |
| | Precipitation | 0.927 | **0.441** | 0.754 | 0.749 | **0.276** | **0.468** | 0.808 | **0.363** | **0.431** | 0.724 | 0.921 | **0.523** |
| | Evapotranspiration | 0.863 | **0.436** | 0.800 | **0.585** | **0.697** | **0.621** | **0.522** | **0.483** | **0.151** | **0.566** | **0.585** | **0.627** |
| | Radiation | 0.735 | 0.900 | 0.918 | 0.885 | 0.836 | 0.901 | 0.880 | 0.925 | 0.912 | 0.922 | 0.713 | 0.828 |
| | Wind speed | 0.946 | 0.963 | 0.932 | 0.943 | 0.867 | 0.911 | 0.860 | 0.731 | 0.917 | 0.917 | 0.933 | 0.929 |
| Suhopolje | Temperature | 0.873 | 0.908 | 0.977 | 0.969 | 0.904 | 0.888 | 0.881 | 0.890 | 0.961 | 0.972 | 0.970 | 0.963 |
| | Precipitation | 0.935 | 0.726 | **0.629** | **0.667** | **0.370** | **0.631** | **0.600** | **0.559** | 0.890 | **0.433** | **0.581** | **0.661** |
| | Evapotranspiration | 0.884 | **0.550** | 0.792 | 0.880 | **0.665** | 0.881 | 0.771 | 0.901 | **0.625** | 0.788 | 0.780 | **0.563** |
| | Radiation | 0.783 | 0.857 | 0.902 | 0.809 | 0.884 | 0.935 | 0.920 | 0.943 | 0.931 | 0.969 | 0.779 | 0.779 |
| | Wind speed | 0.897 | 0.946 | 0.928 | 0.885 | 0.788 | 0.947 | 0.850 | 0.707 | 0.820 | 0.918 | 0.839 | 0.867 |
| Skenderovci | Temperature | 0.720 | 0.877 | 0.976 | 0.983 | 0.941 | 0.916 | 0.947 | 0.893 | 0.970 | 0.938 | 0.962 | 0.962 |
| | Precipitation | 0.813 | 0.811 | **0.646** | 0.952 | **0.484** | **0.524** | 0.865 | **0.335** | **0.549** | **0.338** | **0.599** | **0.572** |
| | Evapotranspiration | **0.669** | **0.425** | **0.681** | 0.817 | 0.775 | 0.896 | 0.785 | 0.921 | **0.670** | **0.684** | 0.815 | **0.442** |
| | Radiation | 0.740 | 0.887 | 0.888 | 0.891 | 0.868 | 0.916 | 0.865 | 0.951 | 0.931 | 0.939 | 0.810 | 0.809 |
| | Wind speed | 0.928 | 0.947 | 0.938 | 0.912 | 0.898 | 0.957 | 0.732 | 0.595 | 0.828 | 0.810 | 0.893 | 0.910 |
| Nedelišće | Temperature | 0.797 | 0.884 | 0.960 | 0.951 | 0.834 | 0.913 | 0.823 | 0.848 | 0.909 | 0.960 | 0.966 | 0.833 |
| | Precipitation | 0.861 | 0.978 | **0.522** | **0.147** | **−0.042** | 0.703 | **0.266** | **0.504** | **0.683** | 0.854 | 0.802 | **0.528** |
| | Evapotranspiration | **0.575** | **0.504** | **0.657** | **0.673** | **0.483** | 0.922 | 0.802 | 0.735 | 0.702 | **0.559** | 0.719 | **0.525** |
| | Radiation | 0.830 | 0.812 | 0.898 | 0.781 | 0.810 | 0.967 | 0.898 | 0.825 | 0.873 | 0.934 | 0.832 | 0.706 |
| | Wind speed | 0.715 | 0.942 | 0.975 | 0.896 | 0.889 | 0.792 | 0.767 | 0.639 | 0.856 | 0.817 | 0.706 | 0.808 |
| Funtane | Temperature | 0.721 | 0.751 | 0.878 | 0.852 | 0.766 | 0.924 | 0.877 | 0.852 | 0.947 | 0.855 | 0.899 | 0.907 |
| | Precipitation | 0.741 | **0.212** | **0.645** | **0.516** | **0.272** | **0.181** | **0.167** | **0.621** | **0.558** | 0.605 | 0.997 | 0.876 |
| | Evapotranspiration | **0.549** | 0.840 | **0.595** | **0.626** | 0.781 | **0.403** | **0.541** | **0.206** | **−0.215** | **0.529** | 0.915 | **0.466** |
| | Radiation | 0.939 | 0.876 | 0.907 | 0.972 | 0.891 | 0.871 | 0.833 | 0.922 | 0.954 | 0.894 | 0.814 | 0.907 |
| | Wind speed | 0.730 | 0.700 | 0.900 | 0.824 | **0.449** | **0.607** | 0.883 | 0.872 | 0.833 | 0.813 | 0.910 | 0.885 |
| Otočac | Temperature | **0.692** | 0.857 | 0.957 | 0.861 | **0.622** | 0.778 | 0.859 | 0.802 | 0.822 | 0.885 | 0.779 | 0.729 |
| | Precipitation | 0.928 | **0.424** | **0.452** | 0.729 | **0.476** | **0.657** | **0.259** | **0.688** | 0.750 | **0.611** | **0.650** | **0.627** |
| | Evapotranspiration | **0.313** | **0.506** | 0.743 | 0.717 | **0.666** | 0.825 | **0.653** | 0.748 | 0.816 | 0.713 | **0.501** | **0.128** |
| | Radiation | **0.639** | 0.938 | 0.881 | 0.942 | 0.843 | 0.824 | 0.807 | 0.829 | 0.879 | 0.890 | **0.660** | **0.288** |
| | Wind speed | 0.828 | 0.824 | 0.915 | 0.797 | 0.752 | 0.762 | 0.864 | 0.775 | 0.805 | 0.798 | 0.889 | **0.650** |
| Oklaj | Temperature | **0.630** | 0.813 | 0.913 | 0.839 | 0.867 | 0.855 | 0.826 | 0.721 | 0.931 | 0.940 | 0.837 | 0.832 |
| | Precipitation | 0.818 | **0.672** | **0.629** | 0.830 | **0.428** | 0.958 | **0.382** | **0.649** | 0.830 | **0.564** | **0.324** | 0.897 |
| | Evapotranspiration | 0.815 | 0.700 | 0.716 | **0.661** | 0.768 | 0.707 | **0.537** | **0.267** | 0.633 | 0.690 | **0.654** | **0.620** |
| | Radiation | 0.855 | 0.927 | 0.910 | 0.935 | 0.915 | 0.912 | 0.829 | 0.778 | 0.936 | 0.949 | 0.866 | 0.830 |
| | Wind speed | 0.904 | 0.932 | 0.905 | 0.826 | 0.822 | 0.855 | 0.939 | 0.776 | 0.809 | 0.940 | 0.925 | 0.917 |
| Potomje | Temperature | 0.844 | 0.893 | 0.935 | 0.914 | 0.942 | 0.930 | 0.911 | 0.803 | 0.939 | 0.957 | 0.953 | 0.853 |
| | Precipitation | 0.907 | **0.578** | 0.831 | 0.864 | **0.392** | 0.745 | **−0.018** | **0.620** | 0.858 | **0.371** | **0.514** | 0.852 |
| | Evapotranspiration | **0.508** | 0.745 | **0.570** | **0.502** | **0.624** | **0.361** | **0.260** | **−0.109** | **−0.580** | 0.804 | **0.442** | **0.699** |
| | Radiation | 0.821 | 0.822 | 0.841 | 0.942 | 0.826 | 0.842 | 0.754 | 0.835 | 0.851 | 0.882 | 0.923 | 0.877 |
| | Wind speed | **0.677** | 0.863 | 0.921 | 0.869 | **0.619** | 0.908 | 0.918 | 0.853 | 0.822 | 0.902 | 0.845 | 0.975 |

**Table A5.** 2021 Correlations between ERA5-Land and Agri4Cast data. Correlations below the correlation threshold value of r = 0.7 are marked in bold. The month with the strongest correlation value for each location is highlighted in orange, the weakest in blue.

| Station | Parameter | Jan | Feb | Mar | Apr | May | Jun | Jul | Aug | Sep | Oct | Nov | Dec |
|---|---|---|---|---|---|---|---|---|---|---|---|---|---|
| Našice | Temperature | 0.973 | 0.976 | 0.968 | 0.972 | 0.856 | 0.974 | 0.931 | 0.972 | 0.943 | 0.975 | 0.912 | 0.953 |
| | Precipitation | 0.847 | 0.812 | 0.738 | **0.379** | **0.634** | **0.678** | **0.618** | **0.481** | **0.290** | 0.851 | **0.579** | **0.468** |
| | Evapotranspiration | **0.518** | 0.750 | **0.600** | 0.849 | 0.847 | **0.484** | **0.273** | 0.802 | 0.720 | **0.406** | 0.735 | **0.696** |
| | Radiation | 0.766 | 0.913 | 0.864 | 0.959 | 0.927 | **0.589** | 0.831 | 0.901 | 0.947 | 0.942 | **0.676** | 0.922 |
| | Wind speed | 0.917 | 0.939 | 0.965 | 0.935 | 0.901 | 0.936 | **0.631** | 0.767 | 0.901 | 0.946 | 0.844 | 0.912 |
| Kamenac | Temperature | 0.966 | 0.985 | 0.971 | 0.978 | 0.881 | 0.979 | 0.930 | 0.972 | 0.959 | 0.970 | 0.901 | 0.958 |
| | Precipitation | 0.802 | 0.756 | 0.801 | **0.421** | 0.713 | **0.555** | **0.350** | **0.367** | **0.639** | 0.847 | 0.731 | **0.669** |
| | Evapotranspiration | **0.611** | **0.694** | **0.503** | 0.897 | 0.871 | **−0.240** | **−0.197** | **0.469** | **0.441** | **0.454** | 0.726 | 0.719 |
| | Radiation | 0.810 | 0.891 | 0.857 | 0.932 | 0.950 | **0.489** | 0.876 | 0.897 | 0.965 | 0.958 | **0.637** | 0.876 |
| | Wind speed | 0.903 | 0.948 | 0.955 | 0.915 | 0.929 | 0.888 | 0.831 | 0.850 | 0.928 | 0.950 | 0.925 | 0.952 |
| Suhopolje | Temperature | 0.971 | 0.981 | 0.958 | 0.972 | 0.842 | 0.968 | 0.857 | 0.981 | 0.935 | 0.981 | 0.920 | 0.942 |
| | Precipitation | **0.780** | 0.904 | 0.915 | **0.450** | **0.503** | 0.835 | **0.606** | **0.556** | **0.319** | **0.610** | **0.548** | **0.490** |
| | Evapotranspiration | **0.671** | **0.785** | **0.644** | 0.899 | 0.912 | **0.437** | **0.420** | 0.803 | 0.705 | **0.606** | 0.742 | **0.680** |
| | Radiation | 0.822 | 0.895 | 0.815 | 0.964 | 0.923 | **0.639** | 0.802 | 0.876 | 0.945 | 0.925 | **0.682** | 0.849 |
| | Wind speed | 0.867 | 0.900 | 0.945 | 0.943 | 0.910 | 0.843 | **0.686** | 0.741 | 0.830 | 0.938 | 0.771 | 0.935 |
| Skenderovci | Temperature | 0.966 | 0.979 | 0.958 | 0.985 | 0.893 | 0.973 | 0.930 | 0.986 | 0.977 | 0.979 | 0.941 | 0.942 |
| | Precipitation | 0.881 | 0.842 | **0.687** | **0.410** | **0.549** | 0.853 | 0.707 | **0.543** | **0.152** | 0.813 | 0.725 | **0.531** |
| | Evapotranspiration | **0.171** | **0.690** | **0.629** | 0.843 | 0.868 | 0.721 | **0.558** | 0.896 | 0.878 | **0.656** | **0.672** | **0.636** |
| | Radiation | 0.713 | 0.917 | 0.780 | 0.958 | 0.883 | 0.784 | 0.808 | 0.898 | 0.954 | 0.923 | **0.626** | 0.914 |
| | Wind speed | 0.862 | 0.869 | 0.959 | 0.927 | 0.883 | 0.827 | **0.592** | **0.326** | **0.683** | 0.874 | 0.812 | 0.865 |
| Nedelišće | Temperature | 0.973 | 0.970 | 0.946 | 0.955 | 0.842 | 0.949 | 0.891 | 0.958 | 0.853 | 0.963 | 0.927 | 0.946 |
| | Precipitation | **0.318** | **0.695** | 0.925 | **0.561** | **0.524** | 0.902 | **0.353** | **0.625** | **0.358** | 0.762 | **0.315** | **0.543** |
| | Evapotranspiration | **0.567** | **0.589** | 0.766 | **0.669** | 0.927 | 0.749 | 0.737 | 0.792 | 0.726 | **0.648** | 0.776 | **0.490** |
| | Radiation | 0.753 | 0.936 | 0.838 | 0.938 | 0.940 | 0.902 | 0.892 | 0.932 | 0.952 | 0.956 | 0.791 | 0.781 |
| | Wind speed | 0.940 | 0.875 | 0.903 | 0.940 | 0.870 | 0.847 | 0.709 | 0.881 | 0.819 | 0.947 | 0.852 | 0.876 |
| Funtane | Temperature | 0.829 | 0.950 | 0.884 | 0.872 | 0.517 | 0.933 | 0.825 | 0.946 | 0.844 | 0.876 | 0.837 | **0.660** |
| | Precipitation | 0.751 | 0.790 | 0.943 | **0.513** | **0.550** | **0.681** | 0.743 | **0.397** | **0.343** | 0.706 | **0.464** | **0.618** |
| | Evapotranspiration | **0.473** | **0.572** | 0.729 | **0.163** | **0.601** | **−0.114** | **0.128** | **0.140** | **−0.125** | 0.823 | **0.491** | **0.453** |
| | Radiation | 0.916 | 0.951 | 0.951 | 0.939 | 0.924 | 0.706 | 0.821 | 0.893 | 0.923 | 0.951 | 0.836 | 0.908 |
| | Wind speed | 0.764 | 0.881 | 0.707 | **0.528** | **0.441** | **0.612** | **0.591** | 0.756 | **0.613** | 0.861 | 0.754 | **0.687** |
| Otočac | Temperature | 0.891 | 0.974 | 0.896 | 0.895 | 0.595 | 0.907 | 0.806 | 0.903 | 0.755 | 0.914 | 0.812 | 0.801 |
| | Precipitation | 0.788 | 0.850 | 0.791 | **0.575** | **0.700** | 0.700 | **0.692** | **0.593** | **0.596** | 0.829 | **0.308** | 0.706 |
| | Evapotranspiration | **−0.045** | 0.714 | 0.795 | 0.840 | 0.787 | **0.501** | 0.759 | 0.846 | 0.785 | **0.405** | **0.573** | **0.436** |
| | Radiation | 0.788 | 0.850 | 0.791 | **0.575** | 0.700 | 0.700 | **0.692** | **0.593** | **0.596** | 0.829 | **0.308** | 0.706 |
| | Wind speed | **0.641** | 0.895 | 0.908 | **0.604** | **0.490** | 0.748 | **0.611** | **0.635** | 0.715 | 0.799 | 0.743 | 0.711 |
| Oklaj | Temperature | 0.944 | 0.967 | 0.936 | 0.959 | 0.798 | 0.930 | 0.710 | 0.945 | 0.896 | 0.888 | 0.872 | 0.810 |
| | Precipitation | 0.821 | **0.666** | **0.613** | **0.569** | **0.214** | 0.872 | **0.442** | **0.490** | 0.856 | **0.624** | **0.507** | **0.555** |
| | Evapotranspiration | **0.642** | 0.714 | 0.745 | 0.873 | 0.815 | **0.308** | **0.116** | **−0.335** | **0.265** | **0.339** | **0.478** | **0.542** |
| | Radiation | 0.875 | 0.940 | 0.918 | 0.851 | 0.849 | 0.706 | 0.902 | 0.862 | 0.928 | 0.928 | 0.895 | 0.863 |
| | Wind speed | 0.905 | 0.943 | 0.944 | 0.930 | 0.923 | 0.790 | 0.834 | **0.676** | 0.913 | 0.905 | 0.724 | 0.844 |
| Potomje | Temperature | 0.976 | 0.950 | 0.877 | 0.969 | 0.910 | 0.992 | 0.875 | 0.936 | 0.920 | 0.958 | 0.866 | 0.895 |
| | Precipitation | 0.851 | **0.628** | 0.865 | **0.409** | 0.762 | **−0.036** | 0.309 | 0.055 | 0.260 | 0.729 | **0.268** | **0.517** |
| | Evapotranspiration | **0.153** | **0.582** | 0.720 | **0.573** | **0.604** | 0.078 | **−0.255** | **−0.124** | 0.485 | 0.030 | 0.489 | 0.095 |
| | Radiation | 0.887 | 0.972 | 0.941 | 0.886 | 0.739 | **0.697** | 0.837 | 0.883 | 0.865 | 0.934 | 0.881 | 0.907 |
| | Wind speed | 0.749 | 0.924 | 0.819 | 0.891 | 0.911 | 0.847 | 0.847 | 0.749 | 0.772 | 0.692 | 0.906 | 0.751 |

# Appendix B

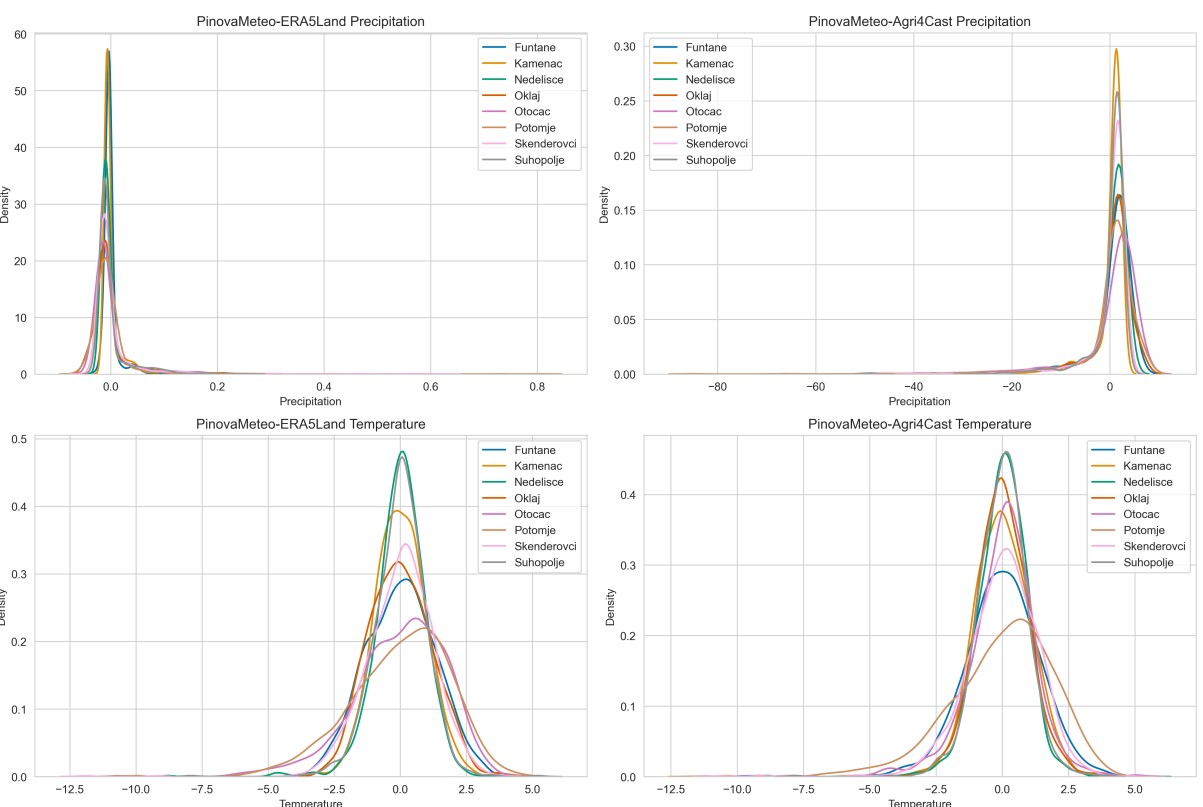

**Figure A1.** Kernel daily density estimation graphs between different data sources.

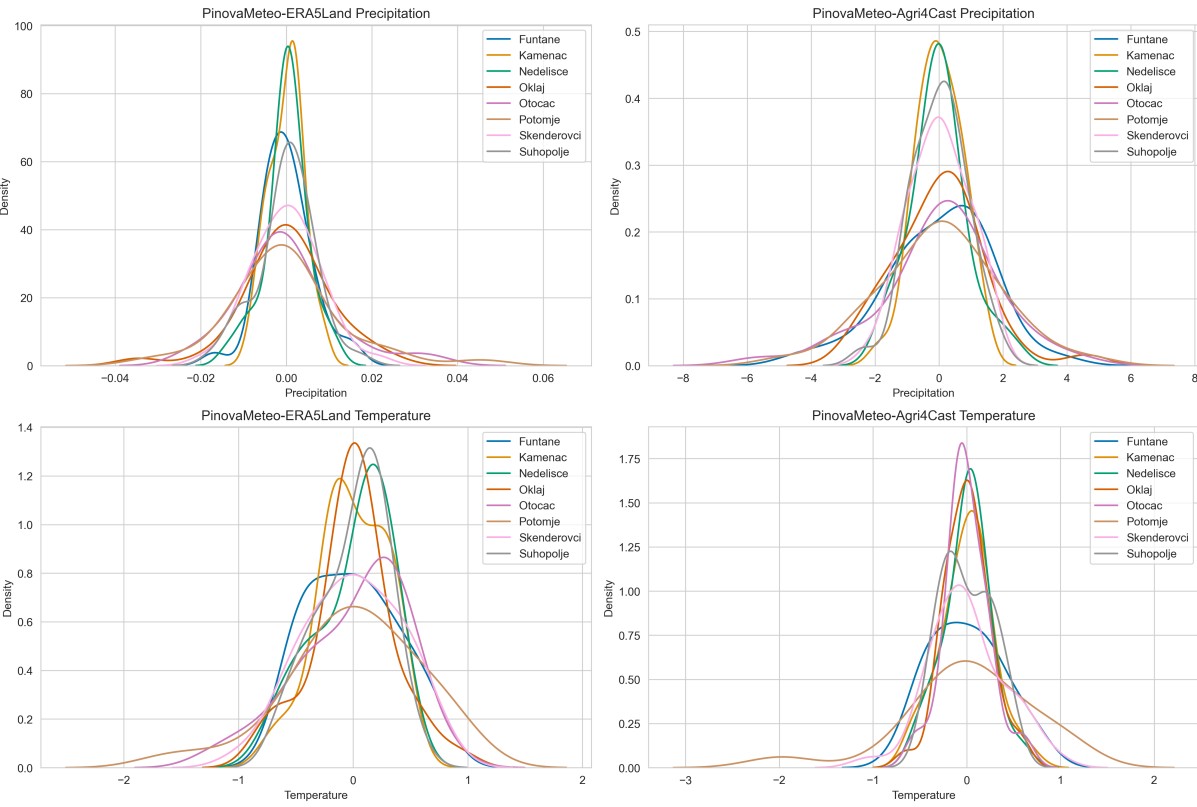

**Figure A2.** Kernel monthly density estimation graphs between different data sources.

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
