# Peer review of "Comparing Remote and Proximal Sensing of Agrometeorological Parameters across Different Agricultural Regions in Croatia: A Case Study Using ERA5-Land, Agri4Cast, and In Situ Stations during the Period 2019–2021"

_remotesensing, doi:10.3390/rs16040641_

Round 1

Reviewer 1 Report

Comments and Suggestions for Authors

The manuscript by Krekovic et al. is devoted to comparison between proximal ground-based and remote satellite-based sensing for precision agriculture in Croatia. It is potentially interesting work to contribute to development of methods of the remote sensing of agrometorological parameters and to increase accuracy thier estimation on basis of the remote satellite-based sensing.

However, I have comments and questions:

1. Title does not seem to be optimal because authors investigate spefic problem; many parameters (e.g., vegetation indices), which can be also usefull for the precision agriculure, are not investigated. I suppose Title should be more specific; e.g., "Comparison of Proximal Ground-Based and Remote Satellite-Based Sensing of Agrometeorological Parameters in Croatia" or similar.

2. Related Work: Measuring vegetation reflectance indices is very improtant way of the remote sensing of agricultural plants. These indices should be discussed in more detail. Particularly, it is interesting: Can thes indices be used as indirect indicator of agrometeorological parameters.

3. Section "3.1. Coprenicus Climate Data Store and ERA5-Land Dataset": The section is not fully clear. Using satellite-based and ground-based investigation seem to be confused. It should be clarified. Particularly, it is not clear: Were paremeters in Table 1 measured by satellite or by ground station? If satellite was used, what optical parameters were used for estimation of precipitation?

4. Section " 3.2. Agri4Cast Portal": Methods interpolation and criteria of their selections should be clarified. 

5. Why was the Pearson correlation coefficient used for estimation of relations? It seems to have low sensetivity to non-linear relations.

Reviewer 2 Report

Comments and Suggestions for Authors

I can understand that this article requires a considerable amount of work. Nevertheless, I still feel uncomfortable with the writing logic of this article, and cannot say that it cannot be done this, but there are indeed some differences in structure and form from many articles.

What I can't accept is that these are using open datasets, and the author only compared and analyzed the relationship between these two datasets. Although this approach has some significance, I think it lacks the author's original work. The data analysis methods used by the author in the article are common and difficult to gain more insights. I think the author should rethink whether more and updated methods can be used to gain more knowledge, rather than simple comparisons.

Overall, the author should delve deeper into the data.

Comments on the Quality of English Language

I think the author's English language is appropriate.

Reviewer 3 Report

Comments and Suggestions for Authors

Precision agriculture is a modern farming approach that leverages technology and data to optimize various aspects of crop and livestock production. One crucial component of precision agriculture is monitoring and sensing, which helps farmers make informed decisions about their farming operations. Two common methods for collecting agricultural data are proximal ground-based sensing and remote satellite-based sensing. This article elaborates on the comparison between these two approaches.

This article is well written with accurate and relevant content. However, I think it should be submitted as a review not an original article. In this paper, Authors compare data from sources that provide freely available meteorological data. There was no original data generated to compare with the meteorological ones.

Suggested title: "Review of Proximal Ground-Based and Remote Satellite-Based Sensing in Precision Agriculture"

Please add bar scale to Figure 1.

Table 4 shows a lot of results. Perhaps the ones with the highest and lowest numbers could be highlighted. Table 5, 6, A3, A4, A5 could be adjusted the same way.

Figure 3 is impossible to read. There might be some patterns, but all the data are overlapping, and it is hard to make any conclusion. My suggestion is to generate the same charts but for each station separately. The most interesting data/findings can be added to results and the rest as an Appendix. Please, consider readjusting Figure A1 and A2 the same way.

Round 2

Reviewer 1 Report

Comments and Suggestions for Authors

Authors considered my comments. I have not additional remarks or questions. The work seems to be interesting and perspective

Author Response

We would like to thank the reviewers for their valuable comments and suggestions, which helped us greatly in improving the paper. 

Reviewer 2 Report

Comments and Suggestions for Authors

The author has already answered my concerns.

Author Response

(The authors gave the same response as above.)

Reviewer 3 Report

Comments and Suggestions for Authors

An original research article is written by the person or people that conducted the experiment or observations. Original research articles are considered empirical or primary sources and present an original study. Articles that look at multiple studies are not considered original research articles. Therefore, this article can not be submitted as an original article. It can be submitted as a review only.

Comments on the Quality of English Language

The quality of English is good.
